# DiffImp: Efficient Diffusion Model for Probabilistic Time Series Imputation with Bidirectional Mamba Backbone

## Abstract

Probabilistic time series imputation has been widely applied in real-world scenarios due to its ability to estimate uncertainty of imputation results. Meanwhile, denoising diffusion probabilistic models (DDPMs) have achieved great success in probabilistic time series imputation tasks with its power to model complex distributions. However, current DDPM-based probabilistic time series imputation methodologies are confronted with two types of challenges: 1) *The backbone modules of the denoising parts are not capable of achieving sequence modeling with low time complexity.* 2) *The architecture of denoising modules can not handle the inter-variable and bidirectional dependencies in the time series imputation problem effectively.* To address the first challenge, we integrate the computational efficient state space model, namely Mamba, as the backbone denosing module for DDPMs. To tackle the second challenge, we carefully devise several SSM-based blocks for bidirectional modeling and inter-variable relation understanding. Experimental results demonstrate that our approach can achieve state-of-the-art time series imputation results on multiple datasets, different missing scenarios and missing ratios.

## 1 Introduction

The analysis of time series can model the intrinsic patterns within time-series data, thus providing robust support for decision-making in various fields, such as meteorology McGovern et al. (2011); Karevan & Suykens (2020), financial analysis Xiang et al. (2022); Owusu et al. (2023); Bai et al. (2020), healthcare Morid et al. (2023); Poyraz & Marttinen (2023) and power systems Tzelepi et al. (2023); Zhou et al. (2021). To enhance the reliability of analytical outcomes, it is critical to ensure the integrity of time series. However, due to various reasons such as device failures, human errors, and privacy protection, time series data can easily be incomplete with missing observations at different timestamps.

Time series imputation methods aim to estimate the values of missing points based on the observed points in incomplete time series, thereby restoring the integrity of the time series while preserving its original statistical properties. According to the ability to provide uncertainty of estimations, time series imputation methods can be categorized into the following two perspectives: 1) *Deterministic* Cao et al. (2018); Cini et al. (2022); Du et al. (2023), and 2) *Probabilistic* Chen et al. (2023b); Kim et al. (2023); Luo et al. (2018) imputation methods. *Probabilistic* time series imputation is particularly important in dealing with complex and uncertain data environments, as it provides a quantification of uncertainty for the imputations. The key to probabilistic imputation lies in modeling the posterior distribution. Existing probabilistic time series imputation methods include *Gaussian Process and Variational Autoencoder*-based methods Fortuin et al. (2020), *Normalization Flow*-based methods Rasul et al. (2021), and *Diffusion*-based methods Tashiro et al. (2021). Among these, the *Diffusion*-based method has emerged as the optimal choice for probabilistic time series due to their accuracy in posterior modeling and adaptability to different scenarios and various types of time series data.

When selecting a denoising backbone in the diffusion model, the following two key factors need to be considered: 1) **Model compatibility**, and 2) **Time complexity**. Model compatibility involves

two key aspects: 1) the backbone of the model should be capable of handling input data effectively. 2) the backbone of the model should align with the model's intended objective (*i.e.*, in diffusion models, the backbone must be capable of modeling noise in the diffusion process). Specifically, the missing observations in time series have correlations with their neighbors on both sides, *so it is crucial to design a model by considering information from neighbors of both sides*. Moreover, *it is also essential to accurately capture the properties of time series*, such as global dependencies and channel correlations. Three mainstream denoising backbones are widely used in diffusion models for time series imputation: 1) *Convoluational Neural Networks (CNNs)-*, 2) *Transformer-* and 3) *State-Space Model (SSM)*-based backbones. Given a time series with a length of $L$, *the CNNs-based backbone* can capture partial information from the neighbors within the receptive fields and has $\mathcal{O}(L)$ time complexity. *The transformer-based backbone* can model temporal dependencies across the entire time series but is with quadratic time complexity $\mathcal{O}(L^2)$. *The SSM backbone* has a linear time complexity, $\mathcal{O}(L)$, but it falls short in capturing the information from one side of the neighbor. Moreover, all these backbones fail to capture the channel dependencies in time series. The comparison results of existing backbones and our method in terms of various dependencies and time complexity are presented in Table.1.

Table 1: Comparison of our method and existing methods in modeling dependencies and time complexity. The results show that our method achieves the most comprehensive data modeling with the lowest time complexity.

| Backbone Model | Global Dependency | Time Complexity | Channel Dependency | Inter-sequence Dependency |
|---|---|---|---|---|
| CNN | Local | $\mathcal{O}(L)$ | Independent | Unidirectional |
| Transformer | Global | $\mathcal{O}(L^2)$ | Independent | Unidirectional |
| SSM | Partial | $\mathcal{O}(L)$ | Independent | Unidirectional |
| DiffImp (Ours) | Global | $\mathcal{O}(L)$ | Dependent | Bidirectional |

In this paper, we propose an efficient diffusion-based framework for probabilistic time series imputation to address the drawbacks in existing backbones of time series imputation, we name it **DiffImp**. To ensure linear complexity, we choose the SSM-based model as the backbone of our framework, which is Mamba Dao & Gu (2024) to be more specific. Though there has been SSM-based diffusion backbones, there remains a question whether the Mamba block is an effective backbone for time series imputation problem and how to design modules suitable for time series imputation problems based on Mamba blocks. To enable Mamba to capture information from both sides of the missing values, we then propose a *Bidirectional Attention Mamba block (BAM)* that is more applicable to time series imputation task. To incorporate bidirectional dependencies, we design a learnable weight module inside the BAM block. This module learns the weights of all points within the sequence, facilitating the modeling of dependencies at different distances.

Next, we propose a *Channel Mamba Block (CMB)* to capture the dependencies among different channels in a time series. Specifically, we treat the variables across different channels in the time series as a sequence of variables and employ the Mamba model alongside the channel dimension, so inter-dependencies among channels can be modeled.

Our contributions are summarized as follows:

- We propose DiffImp, an efficient diffusion-based model for the time series imputation task. It integrates mamba-based blocks as diffusion backbones and equips the model with the capability of probabilistic time series imputation with linear time and space complexity.

- We propose Channel Mamba Block and Bidirectional Attention Mamba block to capture the sequential correlations and channel dependencies inside the time series. The bidirectional attention mamba block and channel mamba block can effectively model the multivariate time series with missing values.

- We conduct experiments on multiple real-world datasets for both time series imputation and time series forecasting tasks. The experimental results demonstrate that our approach achieves state-of-the-art performance across several datasets, different missing scenarios and missing ratios.

## 2 PRELIMINARIES

### 2.1 STATE SPACE MODELS

State Space Models (SSMs) are an emerging approach to model sequential data, which is implemented by finding out state representations to model the relationship between input and output sequences. A SSM receives a one-dimensional sequence $X \in \mathbb{R}^L$ as the input and outputs a corresponding sequence $Y \in \mathbb{R}^M$. Under continuous settings, the SSMs are defined according to Eq.1:

$$\begin{cases} \dot{h}(t) & = \boldsymbol{A}h(t) + \boldsymbol{B}x(t) \\ y(t) & = \boldsymbol{C}h(t) + \boldsymbol{D}x(t), \end{cases} \tag{1}$$

where $x(t) \in \mathbb{R}^L$, $y(t) \in \mathbb{R}^M$, $h(t)$, and $\dot{h}(t) \in \mathbb{R}^N$ stands for the input, output, hidden state, and derivative of hidden state at timestamp $t$, respectively; $\boldsymbol{A} \in \mathbb{R}^{N \times N}, \boldsymbol{B} \in \mathbb{R}^{N \times L}, \boldsymbol{C} \in \mathbb{R}^{M \times N}$ and $\boldsymbol{D} \in \mathbb{R}^{M \times L}$ are learnable model parameters.

In real-world applications, the input sequences are discrete samplings of continuous sequences. According to Gu et al. (2022), under discrete settings, by applying the zero-order hold technique to Eq.1, it can be reformulated as follows.

$$\begin{cases} h_k = \bar{\boldsymbol{A}}h_{k-1} + \bar{\boldsymbol{B}}x_k \\ y_k = \boldsymbol{C}h_k \end{cases}, \tag{2}$$

where $\bar{\boldsymbol{A}} = \exp(\Delta\boldsymbol{A}), \bar{\boldsymbol{B}} = (\Delta\boldsymbol{A})^{-1}(\exp(\Delta\boldsymbol{A}) - \boldsymbol{I}) \cdot (\Delta\boldsymbol{B})$ and $\Delta$ is the learnable step size in discrete sampling. We can see from Eq.2 that the hidden state is updated according to the input $x(t)$ and last hidden state $h(t-1)$ while the output is generated by the hidden state $h(t)$ and the input $x(t)$ and in Gu et al. (2020), where it introduces High-order Polynomial Projection Operator (Hippo) to achieve longer sequence modeling.

However, it is worth noticing that $\boldsymbol{A}, \boldsymbol{B}, \boldsymbol{C}, \boldsymbol{D}$ in Eq.1 and Eq.2 are time-invariant parameters, *i.e.*, they are data-independent parameters and do not change over time. Therefore the model is not capable of assigning different weights at different positions in the input sequence while receiving new inputs. To address this issue, Gu & Dao (2023) proposed Mamba, in which the parameter matrices $\boldsymbol{A}, \boldsymbol{B}, \boldsymbol{C}, \boldsymbol{D}$ are input-dependent, thus enhancing the performance of sequence modeling. To tackle the problem of non-parallelization, Gu & Dao (2023) also introduced selective scan mechanism for effective computing. For further performance and efficiency improvements, Dao & Gu (2024) point out that SSMs can be categorized as a variant of linear attention model. In this work, we follow the same architecture of parallel Mamba Blocks as Dao & Gu (2024) and a RMS-norm Zhang & Sennrich (2019) module is added after the parallel Mamba block. The details of the post-normalization Mamba Block (PNM Block) are illustrated in Fig.3a.

### 2.2 DIFFUSION MODELS

Let $x_t$ be a sequence of variables for $t = 1, 2, \cdots, T$. The diffusion process consists of two processes: 1) **The forward process** without learnable parameters, which transforms the data distribution into a standard Gaussian distribution by gradually adding noise to the data. 2) **The reverse process** with learnable parameters, which first samples from the standard Gaussian distribution and then progressively denoises the data to approximate the data distribution. The reverse process of diffusion models a parameterized distribution $p_\theta$ defined with the following Markov chain to approximate the real data distribution:

$$p_\theta(x_{0:T}) = p(x_T) \prod_{t=1}^{T} p_\theta(x_{t-1}|x_t), \tag{3}$$

where $x_T \sim \mathcal{N}(0, I)$ denotes the latent variable sampled from standard Gaussian distribution and

$$p_\theta(x_{t-1}|x_t) = \mathcal{N}(x_{t-1}; \mu_\theta(x_t, t), \sigma_\theta(x_t, t)I), \tag{4}$$

The loss function of DDPM aims at minimizing the difference between the noise in the forward process $\epsilon$ and the parameterized noise $\epsilon_\theta$ in the reverse process:

$$\mathcal{L}_d = \mathbb{E}_{x_0, \epsilon} \|\epsilon - \epsilon_\theta(x_t, t)\|, \tag{5}$$

where $t$ stands for the diffusion time embedding and $x_t$ is calculated in the forward process. Please refer to Appendix 7.1 for more details about the diffusion models.

## 2.3 PROBLEM FORMULATION

**Definition 1** (**Time Series**). *A time series can be defined as a tuple, denoted as $\tilde{X} = (X, M, T)$, where $X \in \mathbb{R}^{K \times L}$ is the observation matrix with $K$ observations at a time, which are ordered along $L$ time intervals chronologically; $M \in \mathbb{R}^{K \times L}$ is an indicator matrix that indicates whether the observation at $(i, j)$ in $X$ is missing or not: if the observation at position $(i, j)$ is missing, i.e., $X_{i,j} = NA$, then $M_{i,j} = 1$, otherwise, $M_{i,j} = 0$; $T \in \mathbb{R}^L$ is the time stamps of the time series.*

**Definition 2** (**Probabilistic Time Series Imputation**). *Given an incomplete time series $\tilde{X} = (X, M, T)$, where $\sum M < K \cdot L$, the problem of probabilistic time series imputation is to learn an imputation function $\mathcal{M}_\theta$, such that*

$$\bar{X} = \mathcal{M}_\theta(\tilde{X}), \tag{6}$$

*where $\bar{X} \in \mathbb{R}^{K \times L}$ is the imputed time series, where $\bar{X}_{i,j} = \mu_{i,j} \pm \sigma_{i,j}$ denotes the probabilistic output if $M_{i,j} = 1$, otherwise $\bar{X}_{i,j} = X_{i,j}$.*

# 3 METHODOLOGY

## 3.1 DIFFUSION MODELS FOR TIME SERIES IMPUTATION

When dealing with time series imputation using diffusion models, consider a time series $\tilde{X}$, our goal is to model the posterior $P(\bar{X}|X, M, T)$. To make the modeled posterior more precisely, it is natural to introduce conditions to introduce the diffusion process. Considering the short range and long range inter-dependencies within time series, maximizing the observed values utilized in the diffusion process can effectively improve the performance of the imputation results. On the other hand, due to the fact that all the observed values are utilized as condition inputs in the diffusion process, we do not apply any extra process to the observed values to avoid the error accumulation caused by information propagation, the observed values $X_o^c$ are condition inputs for the diffusion process. Thus, the reverse process in Eq.3 is modified to a conditional form with time-series inputs:

$$p_\theta(X_{0:T}^m|X_0, X_o^c) = p(X_T^m) \prod_{t=1}^{T} p_\theta(X_{t-1}^m|X_t^m, X_o^c), \tag{7}$$

where $X_T^m \sim \mathcal{N}(0, I)$, $X_t^m$ denotes the sequence of latent variables in the diffusion process and $t \in \{1, 2, \cdots, T\}$ is the diffusion time steps. Eq.4 is reformulated as:

$$p_\theta(X_{t-1}^m|X_t^m, X_o^c) = \mathcal{N}(X_{t-1}^m; \mu_\theta(X_t^m, t|X_o^c), \sigma_\theta(X_t^m, t|X_o^c)\boldsymbol{I}), \tag{8}$$

the parameterized mean turns to:

$$\mu_\theta(X_t, t) = \frac{1}{\alpha_t}\left(X_t - \frac{\beta_t}{\sqrt{1-\alpha_t}}\epsilon_\theta(X_t, t|X_o^c)\right), \tag{9}$$

where

$$X_t = \sqrt{\alpha_t}X_0 + (1-\alpha_t)\epsilon, \tag{10}$$

and $\{\beta_t \in (0, 1)\}_{t=1}^T$ is a predefined variance scheduler and $\alpha_t = \prod_{i=1}^{t}(1-\beta_t)$, hence we get the conditional diffusion loss for time series imputation task:

$$\mathcal{L} = \mathbb{E}_{X_0, \epsilon}\|\epsilon - \epsilon_\theta(X_t, t|X_o^c)\| = \mathbb{E}_{X_0, \epsilon}\|\epsilon - \epsilon_\theta(\sqrt{\alpha_t}X_0 + (1-\alpha_t)\epsilon, t|X_o^c)\|, \tag{11}$$

where $\epsilon \sim \mathcal{N}(0, I)$.

In the real world, the imputation problem encounters various complexities, such as different ratios of missing data, the positions of missing values within the sequence and the distribution of missing data. To simulate various complex missing situations in real-world scenarios, we adopt a self-supervised approach for training, *i.e.*, applying a predefined mask to the complete dataset to construct corresponding dataset with missing data. We follow the same mask strategies in Alcaraz & Strodthoff (2023), including *Random Missing (RM)* which corresponds to the situation of uniformly random missing values, *Random Block Missing (RBM)* which corresponds to the situation of continuous missing values (missing intervals) in different channels and *Blackout Missing (BM)* which contains missing intervals at the same timestamps among different channels.

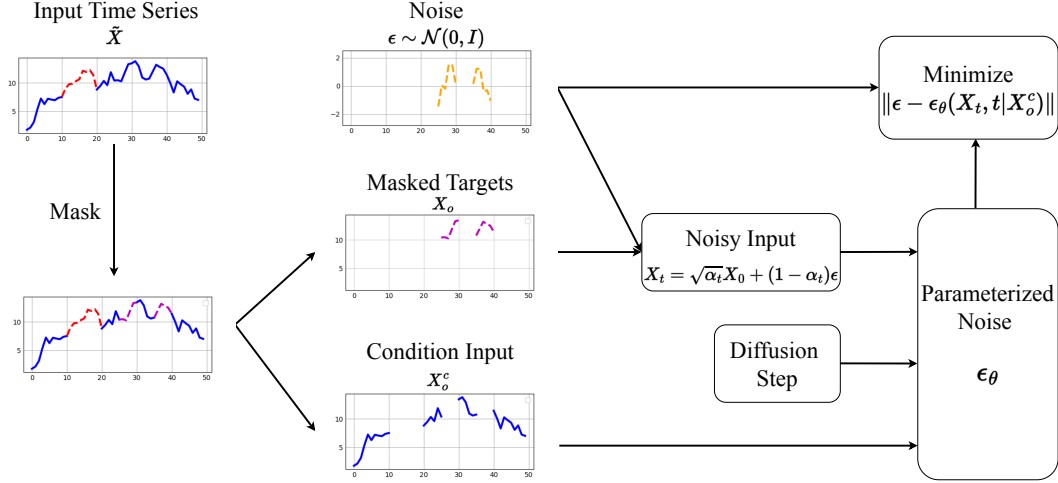

Figure 1: The self-supervised framework and training process of DiffImp. First, some of observed values are masked following the same missing pattern as the missing values (in red) to get masked targets ($X_0$, in magenta) and the condition input ($X_o^c$, in blue). The noisy input is obtain from $X_0$ and $\epsilon$ (in orange) sampled from $\mathcal{N}(0, I)$ The objective of the network is to minimize the difference between the parameterized noise $\epsilon_\theta(X_t, t)$ and $\epsilon$. Solid lines in each time series represent observed values, while dashed lines represent missing values.

## 3.2 MODEL ARCHITECTURE

**The Overall Module Architecture** Fig.1 illustrates the overall self-supervised framework and training process of our model. We first mask part of the observed values according to the pattern of missing values, where the masked values serve as the imputation target $X_0$ during training. The remaining observed values form the conditional input $X_o^c$ for the noise prediction network $\epsilon_\theta$. We then combine $X_0$ with noise $\epsilon$ sampled from a standard normal distribution to obtain the noisy input $X_t$. Both $X_o^c$, $X_t$, and the diffusion step $t$ are fed into the noise prediction network $\epsilon_\theta$ to get the parameterized noise. The network minimizes the difference between $\epsilon_\theta$ and $\epsilon$ according to Eq.11.

As shown in Fig.2, the forward process of $\epsilon_\theta$ are as follows: For each diffusion step, the input consists of the following parts: noisy input $X_t$, the condition input $X_o^c$ and the diffusion step $t$. To begin with, the inputs are embedded to the latent diffusion space. The embedding module of noisy inputs and condition inputs share a similar model structure, which consists of a linear projection module followed by an SMM block in Fig.3b. The SMM block is composed of stacks of Bidirectional Attention Mamba (BAM) blocks and Channel Mamba Blocks (CMB), which is introduced in the next part. Due to the relatively limited information from $t$, the embedding module of $t$ only consists of linear projection modules. After the embedding step, the embedded diffusion step is concatenated with the input embeddings. The concatenated embeddings are fed in to a SMM module. Then the output of the SMM module is concatenated with the condition embeddings. After feeding the final embeddings to another SMM module and final projection module, we can get the noise predictions $\epsilon_\theta(X_t, t)$. The training and sampling algorithm is detailed in Alg.1 and Alg.2.

**Mamba Encoders for Bidirectional Modeling** For probabilistic time series imputation tasks, the objective is to attain a more precise posterior estimation for the missing points contingent upon the observed points. Therefore, our proposed module should achieve two key objectives: Firstly, it should possess bidirectional analysis capability, which means that the model should be able to capture dependencies in both the forward and reverse temporal directions. Secondly, considering that the known points at different positions relative to the missing point have varying distances, the model should assign different weights to difference timestamps. To address these issues, we devise a bidirectional attention Mamba module (BAM). BAM takes the representations from previous layers as input, which are then fed into two distinct PNM modules (Fig.3a), enabling the model to capture bidirectional dependencies. More specifically, temporal attention is implemented by assign-

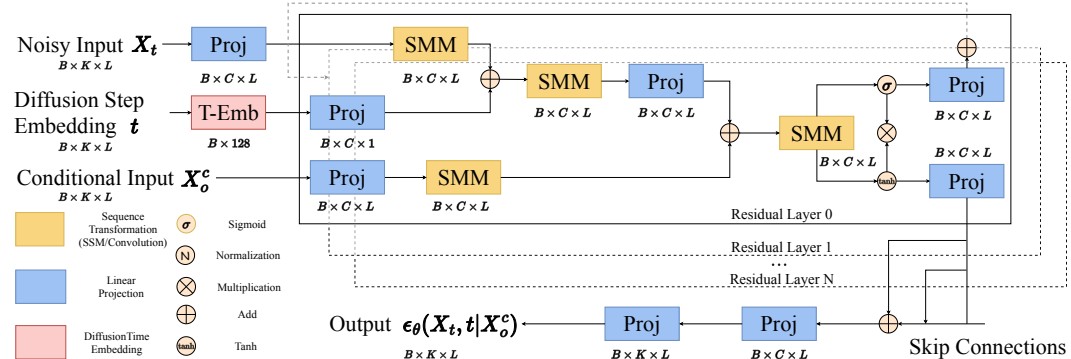

Figure 2: Architecture of $\epsilon_\theta$ in DiffImp

ing different values to various time steps in the sequence, where the temporal attention module also receives the previous layer's representation and learns the weights for different timestamps. The details of BAM are shown in Fig.3d.

**Mamba Encoders for Inter-channel Modeling** In the context of multivariate time series, inter-dependencies exist among variables across different channels. The effective modeling of these inter-channel correlations is instrumental in capturing the intrinsic characteristics of the time series more adeptly. Additionally, when analyzing the relationships between channels, the order of the channels does not exhibit the sequential dependencies as that among timestamps. Consequently, we employ unidirectional channel dependency modelling architecture, termed as Channel Mamba Block (CMB). We first transpose the input time series representation for processing on the channel dimension. The transposed representations are then subjected to a normalization module and processed through a PNM block, yielding a more profound feature representation. The details of CMB are presented in Fig.3c.

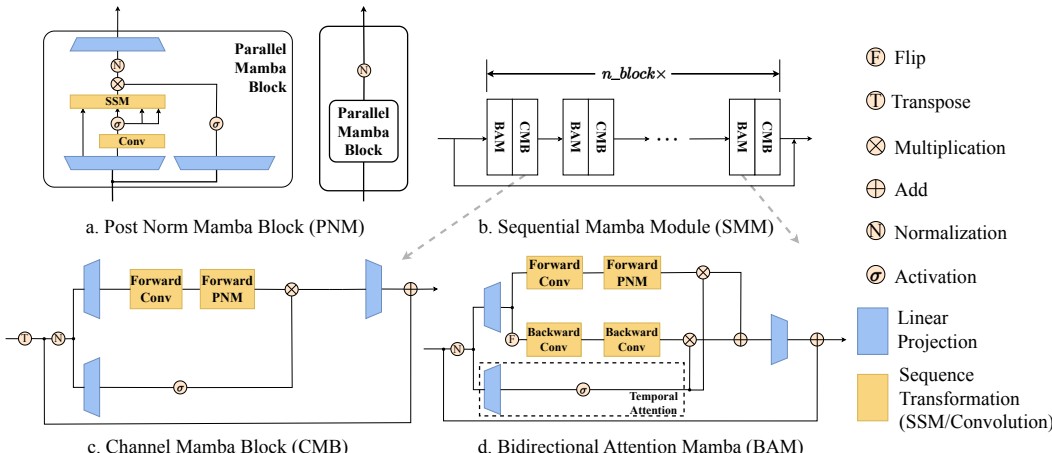

Figure 3: Details of PNM, SMM, CMB, BAM block in the noise prediction module. (a) PNM: backbone module based on Mamba. (b) SMM: core components of noise prediction module, composed of stacks of BAM and CMB. (c) CMB: unidirectional module for inter-channel dependency modeling. (d) BAM: bidirectional module with temporal attention for intra-channel, multi-range dependency modeling.

**Complexity Analysis** While dealing with the input sequences, the core component of our module is the PNM module in Fig.3 and the self-attention module in the Transformer architecture, respectively. In this part, we will give a brief analysis about the time and space complexity in the SSM module and self-attention module[1]. The time complexity of self-attention module is $O(CL^2)$ and the space

---

[1]We do not take the time and space complexity of MLPs before the self-attention module or SSM module into consideration.

---

**Algorithm 1** Training Procedure of DiffImp

---

1: **Input:** Observed sequence $x_0$, number of iterations $N$, variance scheduler $\beta_t$
2: **Output:** Denoising function $\epsilon_\theta$
3: **For** i = 1 **to** $N$ **do**:
4:    $t \sim \text{Uniform}(\{1, 2, \cdots, T\})$
5:    $\epsilon \sim \mathcal{N}(0, I)$
6:    Calculate diffusion targets $x_t$ according to Eq.10
7:    Take gradient step on
$$\nabla_\theta(\|\epsilon - \epsilon_\theta(x_t, t | X_0)\|)$$
   according to Eq.11
8: **End For**

---

**Algorithm 2** Sampling Procedure of DiffImp

---

1: **Input:** Trained denoising function $\epsilon_\theta$, sampling step $T$
2: **Output:** Mean prediction $x_0$
3: **For** $t = T, T - 1, \cdots, 1$ **do**:
4:    $z \sim \mathcal{N}(0, I)$ if $t > 1$ else $z = 0$
5:    $x_{t-1} = \frac{1}{\sqrt{\alpha_t}} \left( x_t - \frac{1-\alpha_t}{\sqrt{1-\bar{\alpha}_t}} \right) \epsilon_\theta(x_t, t) + \sigma_t z$
6: **End For**

---

complexity is $O(L^2 + CL)$, where $L$ is the length of the input sequence and $C$ is the channel of the input sequence.

In our method, the forward process described in Eq.2 is implemented by converting the process to multiplications of structured matrices, which is of time complexity $O(NCL)$ and of space complexity $O(CL + N(C + L))$ ($N$ is a constant number and set as 16 by default). This indicates that our model is of linear time and space complexity with respect to the sequence length $L$, which ensures scalability and reduces memory cost for longer sequences.

## 4 EXPERIMENTS

### 4.1 EXPERIMENT SETTINGS

**Datasets and Experimental Settings** We conduct experiments on five real-world datasets to validate the effectiveness of our approach. These datasets span multiple domains, namely Electricity dataset Asuncion & Newman (2007), MuJoCo dataset Rubanova et al. (2019b), ETTm1 dataset Zhou et al. (2021), Physionet (Healthcare) dataset Silva et al. (2012) and Air quality (AQI) dataset Yi et al. (2016).

All experiments are conducted using PyTorch Paszke et al. (2019) in Python 3.9 and execute on an NVIDIA RTX3090 GPU. The training process is guided by Eq.11, employing the ADAM optimizer Kingma & Ba (2015) with a learning rate of $2 \times 10^{-4}$. More details about the datasets and experimental settings can be found in the Appendix.

**Evaluation Metrics and Baselines** To achieve an extensive evaluation of imputation performance, diverse metrics are utilized for evaluating deterministic imputation results, namely **Mean Absolute Error (MAE)**, **Mean Squared Error (MSE)**, and **Root Mean Square Error (RMSE)**. Due to reproducibility reasons of baselines, we compare with different baselines and report different metrics for different datasets. The datasets and corresponding baseline and metrics are listed in Table.2. We follow the same settings and dataset preprocessings as Alcaraz & Strodthoff (2023) and collect all the baseline results from the same paper.

As for the evaluation of probabilistic imputation, we calculate the **Continuous Ranked Probabilistic Score-sum (CRPS-sum)** on the electricity dataset and **Continuous Ranked Probabilistic Score (CRPS)** on the Physionet dataset and Air quality dataset. The CRPS-sum and CRPS results are collected from Yan et al. (2024). In all the tables of our experiment results, the best results are in

**bold** and second best results are underlined. All the deterministic metrics are maintained by running the experiment for 3 times and CRPS-sum is obtained by 10 runs.

Table 2: Datasets and corresponding evaluation metrics and baselines for time series imputation and forecasting task.

| Dataset | Task | Metric | Baseline |
|---|---|---|---|
| Electricity | Imputation | MAE; RMSE; MRE | M-RNN Yoon et al. (2019); GP-VAE Fortuin et al. (2020); BRITS Cao et al. (2018); SAITS Du et al. (2023); CSDI; SSSD |
| MuJoCo | Imputation | MSE | RNN GRU-D Che et al. (2018); ODE-RNN Rubanova et al. (2019a); NeuralCDE Morrill et al. (2021); Latent-ODE Rubanova et al. (2019a); NAOMI Liu et al. (2019); NRTSI Shan et al. (2023a); CSDI; SSSD |
| Air Quality | Imputation | MAE;MSE | V-RIN Mulyadi et al. (2022);GP-VAE Fortuin et al. (2020);BRITS Cao et al. (2018); SPIN Marisca et al. (2022);SPIN-H Marisca et al. (2022);gatgpt Chen et al. (2023a); GRIN Cini et al. (2022);CSDI |
| | | RMSE | V-RIN Mulyadi et al. (2022);BRITS Cao et al. (2018); SSGAN Miao et al. (2021);RDIS Choi et al. (2023);CSDI;SSSD; CSBI Chen et al. (2023b);TS-diff Kollovieh et al. (2023);SAITS Du et al. (2023); D³M Yan et al. (2024);TIDER Liu et al. (2023) |
| Physionet | Imputation | RMSE | V-RIN Mulyadi et al. (2022);BRITS Cao et al. (2018); SSGAN Miao et al. (2021);RDIS Choi et al. (2023);CSDI;SSSD; CSBI Chen et al. (2023b);TS-diff Kollovieh et al. (2023);SAITS Du et al. (2023); D³M Yan et al. (2024);TIDER Liu et al. (2023) |
| ETTm1 | Forecasting | MAE; MSE | LSTNet Lai et al. (2018); LSTM Bahdanau et al. (2015); Reformer Kitaev et al. (2020); LogTrans Li et al. (2019); Informer Zhou et al. (2021); Autoformer Wu et al. (2021); CSDI; SSSD |

## 4.2 TIME SERIES IMPUTATION

**Deterministic Imputation Results** Table.3 presents the experimental results on the MuJoCo dataset under $RM$ missing scenario with high missing ratios of 70%, 80%, and 90%, respectively. On the MuJoCo dataset, DiffImp achieves SOTA performance under 80% and 90% missing ratio, delivering at least 50% performance improvement over previous SOTA methods. In the experiment of 70% missing ratio, our method achieves results very close to SOTA. The results on MuJoCo dataset indicate that our proposed DiffImp is the optimal method for high missing ratio imputation under the $RM$ missing pattern. Table.4 shows the experimental results on the Electricity dataset, where we apply the $RM$ missing pattern with missing ratios of 10%, 30%, and 50%. We achieve the best results across all metrics with a 30% missing ratio, significantly outperforming other methods. In the experiments with 10% and 50% missing ratios, we obtain results with only a slight gap to the SOTA models.

Table 3: MSE Results on MuJoCo Dataset with missing ratio 70%, 80% and 90% for the missing scenario $RM$.

| Model | 70% RM | 80% RM | 90% RM |
|---|---|---|---|
| RNN GRU-D | 1.134e-2 | 1.421e-2 | 1.968e-2 |
| ODE-RNN | 9.86e-3 | 1.209e-2 | 1.647e-2 |
| NeuralCDE | 8.35e-3 | 1.071e-2 | 1.352e-2 |
| Latent-ODE | 3.00e-3 | 2.95e-3 | 3.60e-3 |
| NAOMI | 1.46e-3 | 2.32e-3 | 4.42e-3 |
| NRTSI | 6.3e-4 | 1.22e-3 | 4.06e-3 |
| CSDI | **2.4e-4±3e-5** | 6.1e-4±1.0e-4 | 4.84e-3±2e-5 |
| SSSD | 5.9e-4±8e-5 | 1e-3±5e-5 | 1.90e-3±3e-5 |
| DiffImp (Ours) | 2.7e-4±1e-5 | **3.16e-4±9.77e-6** | **6.5e-4±1e-4** |

Table 4: MAE and RMSE results on Electricity Dataset

| | 10% RM | | | 30% RM | | | 50% RM | | |
|---|---|---|---|---|---|---|---|---|---|
| Model | MAE | RMSE | MRE | MAE | RMSE | MRE | MAE | RMSE | MRE |
| M-RNN | 1.244 | 1.867 | 66.6% | 1.258 | 1.876 | 67.3% | 1.283 | 1.902 | 68.7% |
| GP-VAE | 1.094 | 1.565 | 58.6% | 1.057 | 1.571 | 56.6% | 1.097 | 1.572 | 58.8% |
| BRITS | 0.847 | 1.322 | 45.3% | 0.943 | 1.435 | 50.4% | 1.037 | 1.538 | 55.5% |
| SAITS | 0.735 | 1.162 | 39.4% | 0.790 | 1.223 | 42.3% | 0.876 | 1.377 | 46.9% |
| CSDI | 1.510±3e-3 | 15.012±4e-2 | 81.10±1e-1% | 0.921±8e-3 | 8.732±7e-2 | 49.27±4e-1% | 0.278±4e-3 | 2.371±3e-2 | 14.93±1e-1% |
| SSSD | **0.345±1e-4** | 0.554±5e-5 | **18.4±5e-3%** | 0.407±5e-4 | 0.625±1e-4 | 21.8±0% | 0.532±1e-4 | **0.821±1e-4** | **28.5±1e-2%** |
| DiffImp (Ours) | 0.378±6e-4 | **0.522±3e-3** | 20.2±1e-2% | **0.348±1e-3** | **0.496±2e-3** | **18.6±1e-1%** | 0.546±3e-3 | 0.837±7e-3 | 29.2±2e-1% |

**Probabilistic Imputation Results** Table.5 presents a comparison of our method with other probabilistic time series imputation methods based on the CRPS-sum metric. The baselines for CRPS-sum include Tashiro et al. (2021); Chen et al. (2023b); Alcaraz & Strodthoff (2023); Yan et al. (2024); Kollovieh et al. (2023). The experimental results show that our method achieves a 21.4% performance improvement compared to the second-best method. This indicates that our method models

Table 5: The CRPS-sum results on electricity dataset

| Model | CSDI | CSBI | SSSD | TS-Diff | D³M | DiffImp(Ours) |
|---|---|---|---|---|---|---|
| CRPS-sum | 2.14e-2±8e-3 | 2.19e-2±7e-3 | 1.96e-2±1e-3 | 2.23e-2±6e-3 | 1.92e-2±4e-3 | **1.51e-2±4e-4** |

the data distribution of the sequence more accurately than other baseline methods. Please refer to the Appendix for more experiment results on probabilistic and determinsitc time series imputation.

## 4.3 TIME SERIES FORECASTING

As mentioned in 3.1, the probabilistic time series forecasting problem can be treated as a variant of the probabilistic time series imputation problem (as a special case of the missing manner *BM*). Therefore, we also conduct experiments to validate the effectiveness of our experiments on probabilistic time series forecasting task. Following the setup in previous works, we test five different forecasting horizons: 24, 48, 96, 288, and 672 time steps, with corresponding conditional lengths (i.e., the length of observed sequence) of 96, 48, 284, 288, and 384 time steps.

Table.6 presents the experimental results on the ETTm1 dataset. Our method achieves state-of-the-art performance on prediction length of 24 and 96, outperforms other imputation-based algorithms at the prediction length of 672, and shows only a slight gap compared to the best imputation-based algorithms at the prediction length of 48 and 288.

Table 6: MSE and MAE results on ETTm1 dataset

| Forecasting Length | 24 | | 48 | | 96 | | 288 | | 672 | |
|---|---|---|---|---|---|---|---|---|---|---|
| Model | MAE | MSE | MAE | MSE | MAE | MSE | MAE | MSE | MAE | MSE |
| LSTNet | 1.170 | 1.968 | 1.215 | 1.999 | 1.542 | 2.762 | 2.076 | 1.257 | 2.941 | 1.917 |
| LSTMa | 0.629 | 0.621 | 0.939 | 1.392 | 0.913 | 1.339 | 1.124 | 1.740 | 1.555 | 2.736 |
| Reformer | 0.607 | 0.724 | 0.777 | 1.098 | 0.945 | 1.433 | 1.094 | 1.820 | 1.232 | 2.187 |
| LogTrans | 0.412 | 0.419 | 0.583 | 0.507 | 0.792 | 0.768 | 1.320 | 1.462 | 1.461 | 1.669 |
| Informer | 0.369 | **0.323** | 0.503 | 0.494 | 0.614 | 0.678 | 0.786 | 1.056 | 0.926 | 1.192 |
| CSDI | 0.370±3e-3 | 0.354±1.5e-2 | 0.546±2e-3 | 0.750±4e-3 | 0.756±1.1e-2 | 1.468±4.7e-2 | 0.530±4e-3 | **0.608±3.5e-2** | 0.891±3.7e-2 | 0.946±5.1e-2 |
| Autoformer | 0.403 | 0.383 | **0.453** | **0.454** | 0.463 | 0.481 | **0.528** | 0.634 | **0.542** | **0.606** |
| SSSD | 0.361±6e-3 | 0.351±9e-3 | 0.479±8e-3 | 0.612±2e-3 | 0.547±1.2e-2 | 0.538±1.3e-2 | 0.648±1.0e-2 | 0.797±5e-3 | 0.783±6.6e-2 | 0.804±4.5e-2 |
| DiffImp (Ours) | **0.282±1.8e-2** | 0.331±9.9e-3 | 0.679±5.6e-3 | 0.548±5.6e-4 | **0.3906±1.3e-2** | **0.4211±8.5e-3** | 0.621±2.1e-3 | 0.741±3.3e-3 | 0.683±3.1e-3 | 0.783±6.8e-3 |

## 4.4 VISUALIZATION RESULTS

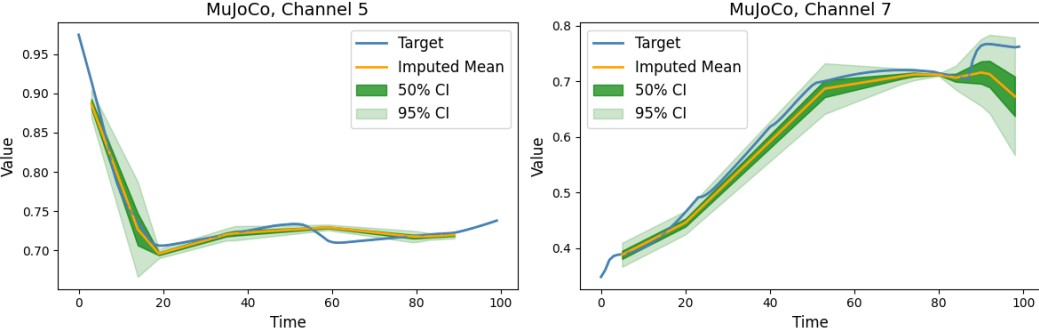

Figure 4: Visualized results of probabilistic time series imputation on MuJoCo dataset.

Fig.4 shows the visualization results for channel 5 and channel 7 on the MuJoCo dataset with a 90% missing ratio. From the figure, we can see that almost all ground truth values for the points to be imputed fall within the 95% confidence interval, and most of the ground truth values are within the 50% confidence interval, which demonstrates the effectiveness of our method. Please refer to the appendix for more visualization results on different datasets.

## 4.5 PARAMETER SENSITIVITY AND SAMPLING TIME ANALYSIS

Table.7 presents the results of parameter sensitivity experiments. In our setup, there are three hyperparameters with different dimensions: sequence dimension, residual connection dimension, and input projection dimension. These three parameters are set to be equal in our experiments. We test different results for $C = 32, 64, 128$. The experimental results show that as $C$ increases, all metrics significantly decrease. Additionally, since $C = 256$ exceeds the single GPU memory capacity, and the performance improvement from $C = 64$ to $C = 128$ is limited, which means the performance improvement by further adding channels may be limited, we choose $C = 128$ in our experiments to balance between metrics and computational cost.

Table.8 presents a comparison of sampling time between our method and other backbone-based methods across different datasets. We find that, with consistent model parameter sizes, our method

Table 7: Parameter sensitivity results.

| #Channel | MAE | MSE | MRE | RMSE |
|---|---|---|---|---|
| 32 | 0.0482±0.0004 | 0.0066±0.0004 | 0.0496±0.00131 | 0.0809±0.0025 |
| 64 | 0.0147±0.00030 | 0.00075±0.00007 | 0.0151±0.00031 | 0.0273±0.0012 |
| 128 | 0.0135±0.00075 | 0.00065±0.00001 | 0.0139±0.00076 | 0.0254±0.0020 |

exhibits inference times similar to the SSSD method with SSM backbone of linear time complexity and CSDI has the shortest inference time due to its CNN backbone. Moreover, as the number of channels increases, the memory consumption of our method increases linearly, indicating that our method demonstrates linear time and space complexity.

Table 8: Model size, inference time and gpu memory cost analysis of CSDI, SSSD and DiffImp on Electricity and MuJoCo dataset.

| | CSDI | | SSSD | | DiffImp (C=64) | | DiffImp (C=96) | | DiffImp (C=128) | |
|---|---|---|---|---|---|---|---|---|---|---|
| | electricity | MuJoCo | electricity | MuJoCo | electricity | MuJoCo | electricity | MuJoCo | electricity | MuJoCo |
| Model size (M) | 2.35 | 0.05 | 49.23 | 48.3 | 24.21 | 24 | 51.03 | 50.92 | 87.7 | 87.57 |
| Inference time(s) | 0.10 | 0.051 | 0.42 | 0.416 | 0.268 | 0.264 | 0.548 | 0.543 | 0.936 | 0.936 |
| GPU Memory Cost (MB) | 4046 | 3226 | 2534 | 2448 | 1662 | 1748 | 2724 | 2696 | 4604 | 4574 |

## 4.6 ABLATION STUDIES

To validate the effectiveness of the proposed module, we conduct ablation experiments on the following aspects: 1) the bidirectional modeling 2) the temporal attention mechanism 3) the inter-channel multivariate dependencies. We also replace the CMB block with channel attention module implemented using Hu et al. (2018) to validate the effectiveness of CMB block. All experiments are conducted on the MuJoCo dataset with the missing ratio 90%. During ablation experiments, we find out that our model converges much slower than other models in the ablation experiment, so we train till all models are converged (for same number of iterations, even if it has already been converged). The hyperparameters in the ablation studies are presented in the appendix.

The results are shown in Table.9. It can be observed that the module equipped with BAM and CMB block performs the best, significantly outperforming the results of removing any one of these components across all four metrics. The temporal attention module has the largest impact on the model, and its removal leads to a significant performance drop. Similarly, removing the CMB module also results in a notable degradation in performance. On the other hand, adjusting the BAM module to its unidirectional form also causes some degree of performance decrease. This fully demonstrates the effectiveness of our proposed blocks.

Table 9: Experimental results of Ablation Study

| Time Modeling | Temporal Attention | Inter-Channel Dependency | MSE | MAE | MRE | RMSE |
|---|---|---|---|---|---|---|
| Bidirectional | Yes | Yes | **5.46e-4±1.6e-5** | **1.17e-2±7.4e-5** | **1.21±7.5e-3%** | **2.33e-2±3.1e-4** |
| Forward | Yes | Yes | 7.19e-4±2.0e-5 | 1.26e-2±2.1e-4 | 1.29±2.2e-2% | 2.67e-2±2.9e-4 |
| Forward | Yes | No | 7.48e-4±9.5e-5 | 1.23e-2±3.5e-4 | 1.23±3.5e-2% | 2.71e-2±1.5e-3 |
| Backward | Yes | Yes | 7.24e-4±7.3e-5 | 1.30e-2±4.2e-4 | 1.30±4.2e-2% | 2.69e-2±1.3e-3 |
| Backward | Yes | No | 8.39e-4±6.1e-5 | 1.46e-2±3.8e-4 | 1.46±3.8e-2% | 2.89e-2±1.0e-3 |
| Bidirectional | Yes | No | 8.85e-4±2.8e-5 | 1.40e-2±3.1e-4 | 1.44±3.4e-2% | 2.97e-2±4.8e-4 |
| Bidirectional | No | Yes | 9.66e-4±9.5e-5 | 1.53e-2±3.4e-4 | 1.57±3.5e-2% | 3.09e-2±1.3e-3 |
| Bidirectional | Yes | Channel Attention | 7.43e-4±4.0e-5 | 1.31e-2±5.5e-5 | 1.35±5.6e-3% | 2.71e-2±5.6e-5 |

## 5 CONCLUSION AND FUTURE WORK

In this paper, we propose DiffImp, a time series imputation model based on DDPM and Mamba backbone, which incorporates bidirectional information flow, temporal attention and inter-variable dependencies. DiffImp enables efficient time series modeling with linear complexity. Experimental results demonstrate that DiffImp achieves superior performance across multiple datasets, various missing patterns, and different missing ratios.

For future work, one possible direction is to further reduce the time complexity of the sampling process while already lowering the complexity of time series modeling, in order to enhance the model's inference efficiency. Another possible direction is to extend the application of diffusion models by applying DiffImp to other time series downstream tasks and time series representation learning tasks.

## 6 REPRODUCIBILITY

To ensure reproducibility and facilitate experimentation, datasets and code are available at: https://anonymous.4open.science/r/DiffImp-843F.

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

## 7 APPENDIX

### 7.1 DETAILS OF DDPM

The denoising diffusion probabilistic model (DDPM) generates unknown data by modeling the distribution of known training data with a parameterized distribution and sampling from the modeled distribution. Concretely, a typical DDPM model consists of two processes, namely the forward process and the reverse process. The forward process of the DDPM model is defined by a Markov chain, which adds noise sampled from standard gaussian noise to initial data distribution $q_0$ step by step until $q_0$ is transformed to standard gaussian distribution $q_T = \mathcal{N}(0, I)$. In every single step, the amount of noise injected to the data distribution at current step is controlled by predefined varaince scheduler $\{\beta_T \in (0,1)\}_{t=1}^T$, which means the injected noise is not learnable. The forward process is defined as follows:

$$q(x_{1:T}|x_0) = \prod_{t=1}^T q(x_t|x_{t-1}), \tag{12}$$

where $x_0, x_1, \cdots, x_t$ stands for the latent variables in the Markov chain and

$$q(x_t|x_{t-1}) = \mathcal{N}(x_t; \sqrt{(1-\beta_t)}x_{t-1}, \beta_t I), \tag{13}$$

Based on Eq.12 and Eq.13, $x_t$ can be represented with a closed form of:

$$x_t = \sqrt{\alpha_t}x_0 + (1-\alpha_t)\epsilon, \tag{14}$$

where $\alpha_t = \prod_{i=1}^t (1-\beta_t)$ and $\epsilon \sim \mathcal{N}(0, I)$.

Correspondingly, the reverse process simulates the denoising of a standard Gaussian distribution $p_t = \mathcal{N}(0, I)$ to the target distribution $p_0$, the entire reverse process is formulated as the following Markov chain:

$$p_\theta(x_{0:T}) = p(x_T) \prod_{t=1}^T p_\theta(x_{t-1}|x_t), \tag{15}$$

where $x_T \sim \mathcal{N}(0, I)$ denotes the latent variable sampled from standard Gaussian distribution and

$$p_\theta(x_{t-1}|x_t) = \mathcal{N}(x_{t-1}; \mu_\theta(x_t, t), \sigma_\theta(x_t, t)I), \tag{16}$$

where $\mu_\theta(x_t, t)$ is parameterized by a neural network and $\sigma_\theta(x_t, t)$ is determined by predefined variance scheduler, *i.e.*:

$$\mu_\theta(x_t, t) = \frac{1}{\alpha_t}\left(x_t - \frac{\beta_t}{\sqrt{1-\alpha_t}}\epsilon_\theta(x_t, t)\right) \tag{17}$$

and

$$\sigma_\theta(x_t, t) = \tilde{\beta}_t^{\frac{1}{2}}, \tag{18}$$

where

$$\tilde{\beta}_t = \begin{cases} \frac{1-\alpha_{t-1}}{1-\alpha_t}\beta_t & t > 1 \\ \beta_1 & t = 1 \end{cases} \tag{19}$$

and $\epsilon_\theta$ is a learnable denoising function.

The loss function of DDPM aims at minimizing the difference between the noise in the forward process $\epsilon$ and the parameterized noise $\epsilon_\theta$ in the reverse process:

$$\mathcal{L}_d = \mathbb{E}_{x_0, \epsilon} \|\epsilon - \epsilon_\theta(x_t, t)\|, \tag{20}$$

where $t$ stands for the diffusion time embedding and $x_t$ is defined in Eq.14.

## 7.2 EXPERIMENT DETAILS

### 7.2.1 DATASET DESCRIPTIONS

In this part, we give a brief introduction about the datasets in our experiments and the details of the datasets are presented in Table.10.

Table 10: Details of MuJoCo, Electricity and ETTm1 dataset

| Dataset | #Train Size | #Test Size | #Sample Length | #Features | #Conditional Values | #Target Values |
|---|---|---|---|---|---|---|
| MuJoCo | 8000 | 2000 | 100 | 14 | 10,20,30 | 90,80,70 |
| Electricity | 817 | 921 | 100 | 370 | 90,70,50 | 10,30,50 |
| ETTm1 | 33865,34417,34000, 33600,33200 | 11490,10000,11420, 10000,10000 | 120,96,480, 576,1056 | 7 | 96,48,384, 288,384 | 24,48,96, 288,672 |

**Air quality**: The air quality dataset contains PM2.5 data from 36 monitor stations in Beijing, which is sampled hourly for 12 months. There are 13.3% of missing values with a non-random missing pattern. The air quality dataset contains artificial ground truth with structured missing pattern.

**Healthcare (Physionet)**: The healthcare dataset contains 4000 irregularly-sampled clinical time series made up of 35 variables (such as Albumin and heart-rate) for 48 hours collected from ICU. To be consistent with previous studies, the dataset is processed hourly to get 48 timesteps and the processed dataset contains near 80% missing values without ground truth. For evaluation, we randomly choose 10/50/90% of the observed values as the ground truth of test dataset.

**PTB-XL:** The PTB-XL ECG dataset consists of 21,837 clinical 12-lead ECGs (i.e. 12 channels) from 18,885 patients, with each ECG lasting 10 seconds at a sampling rate of 100 Hz and the missing ratio is ser as 20%. For the 248 time-step setting, the dataset was preprocessed on crops, corresponding to 69,764 training samples and 8,812 test samples.

**MuJoCo**: The MuJoCo dataset Rubanova et al. (2019b) collects a total of 10,000 simulations of the "Hopper" model from the DeepMind Control Suite and MuJoCo simulator. The position of the body in 2D space is uniformly sampled from the interval $[0, 0.5]$. The relative position of the limbs is sampled from the range $[-2, 2]$, and initial velocities are sampled from the interval $[-5, 5]$. In all, there are 10000 sequences of 100 regularly sampled time points with a feature dimension of 14 and a random split of $80/20$ is done for training and testing. We follow the same preprocessing as in Shan et al. (2023b) for fair comparison.

**Electricity**: The Electricity dataset from the UCI repository Asuncion & Newman (2007) contains electricity usage data (in kWh) collected from 370 clients every 15 minutes. The dataset is collected and preprocessed as described in Du et al. (2023). Since the dataset does not contain missing values, values of the complete dataset are randomly dropped for the computation of targets according to the RM scenario and the data is already normalized. The first 10 months of data (2011/01 - 2011/10) are designated as the test set, the following 10 months of data (2011/11 - 2012/08) as the validation set, and the remaining data (2012/09 - 2014/12) as the training set. The training and test sets are directly utilized, while the validation set is excluded. The dataset comprises 817 samples, each with a length of 100 time steps and the aforementioned 370 features. Specifically, the 370 channels are split into 10 batches of 37 features each. Mini-batches of 43 samples, each containing 37 features and a respective length of 100, are then passed to the network to ensure that no data is dropped during training.

**ETTm1**: This dataset contains the amount of detail required for long-time series forecasting based on the Electricity Transformer Temperature (ETT). The data set contains information from a compilation of 2-year data from two distinct Chinese counties. In our experiment, we work with ETTm1 which covers data at a 15-minute level. The data is composed of the target value oil temperature and six power load features. We follow the same preprocessing as in Zhou et al. (2021) and cover five different forecasting horizons $\{24, 48, 96, 288, 672\}$ with corresponding observed length $\{96, 48, 384, 288, 384\}$.

### 7.2.2 SUPPLEMENTARY EXPERIMENT RESULTS

Table.11 shows our MAE and MSE performance on the Air Quality dataset. We observe that we achieve the best performance on the MAE metric, with a 29.7% improvement compared to the second-best result. For the MAE metric, we achieve the second-best performance, with only a 1.85% difference compared to the best result.Table.12 shows our RMSE performance on the Physionet and Air Quality datasets. We can observe that on the Physionet dataset, for the 10%, 50%, and 90% missing rates, our method achieves the best performance, with improvements of 22.6%, 17.2%, and 23.5% compared to the second-best result. On the Air Quality dataset, our method performs only 2.58% higher than the best method.

Table 11: MAE and MSE results on air quality dataset. - denotes the MSE result is not provided in the original paper.

|  | AQI | |
|---|---|---|
|  | MAE | MSE |
| V-RIN | 25.4±0.62 | - |
| GP-VAE | 25.71 | 2589.53 |
| BRITS | 14.1±0.26 | 495.94±43.56 |
| SPIN | 11.77±0.54 | - |
| SPIN-H | 10.89±0.27 | - |
| gatgpt | 10.28 | **341.26** |
| GRIN | 10.51±0.28 | 371.47±17.38 |
| CSDI | 9.60±0.04 | - |
| DiffImp | **6.75±0.014** | 347.58±0.55 |

Table 12: RMSE results on PhysioNet and Air quality dataset.

|  | Physionet | | | AQI |
|---|---|---|---|---|
|  | 10% missing | 50% missing | 90% missing |  |
| V-RIN | 0.628±0.025 | 0.693±0.022 | 0.928±0.013 | 40.11±1.14 |
| BRITS | 0.619±0.018 | 0.701±0.021 | 0.847±0.021 | 24.28±0.65 |
| SSGAN | 0.607±0.034 | 0.758 ±0.025 | 0.830±0.009 | - |
| RDIS | 0.635±0.018 | 0.747 ±0.013 | 0.922±0.018 | 37.25±0.31 |
| CSDI | 0.531±0.009 | 0.668±0.007 | 0.834±0.006 | 19.21±0.13 |
| CSBI | 0.547±0.019 | 0.649 ±0.009 | 0.837±0.012 | 19.07±0.18 |
| SSSD | 0.459±0.001 | 0.632±0.004 | 0.824±0.003 | 18.77±0.08 |
| TS-Diff | 0.523±0.015 | 0.679±0.009 | 0.845±0.007 | 19.06±0.14 |
| SAITS | 0.461±0.009 | 0.636±0.005 | 0.819±0.002 | 18.68±0.13 |
| D$^3$M | 0.438±0.003 | 0.615±0.012 | 0.814±0.002 | **18.19±0.18** |
| TIDER | 0.486±0.006 | 0.659±0.009 | 0.833±0.005 | 18.94±0.21 |
| DiffImp (Ours) | **0.339±0.0002** | **0.509±0.007** | **0.623±0.0001** | 18.66±0.26 |

Table 13: CRPS results on PhysioNet and Air Quality dataset.

|  | Physionet | | | AQI |
|---|---|---|---|---|
|  | 10% missing | 50% missing | 90% missing |  |
| GP-VAE | 0.582±0.003 | 0.796±0.004 | 0.998±0.001 | 0.402±0.009 |
| V-RIN | 0.814±0.004 | 0.845±0.002 | 0.932±0.001 | 0.534±0.013 |
| CSDI | 0.242±0.001 | 0.336±0.002 | 0.528±0.003 | 0.108±0.001 |
| CSBI | 0.247±0.003 | 0.332 ±0.003 | 0.527±0.006 | 0.110±0.002 |
| SSSD | 0.233±0.001 | 0.331±0.002 | 0.522±0.002 | 0.107±0.001 |
| TS-Diff | 0.249±0.002 | 0.348±0.004 | 0.541±0.006 | 0.118±0.003 |
| D$^3$M | 0.223±0.001 | 0.327±0.003 | **0.520±0.001** | 0.106±0.002 |
| DiffImp (Ours) | **0.164±0.0004** | **0.2438±0.00008** | 0.533±0.0004 | **0.0959±0.0002** |

Table.13 shows our CRPS performance on the Physionet and Air Quality datasets. As shown in the table, we observe that on the Physionet dataset, for the 10% and 50% missing rates, our method

achieves the best performance, with improvements of 35.98% and 25.4% compared to the second-best result. On the Air Quality dataset, our method also achieves the best performance, with a 9.5% improvement over the second-best method.

Table.14 shows our imputation performance on the ECG data (PTB-XL dataset). We can conclude that on the PTB-XL dataset, for a 20% missing rate, under three different missing scenarios—RM, RBM, and BM, our method achieves state-of-the-art performance.

Table 14: MAE and RMSE Results on ECG data (PTB-XL dataset). The best results are in **bold** and second best results are underlined.

| Model | MAE | RMSE |
|---|---|---|
| 20% RM on PTB-XL | | |
| LAMC | 0.0678 | 0.1309 |
| CSDI | 0.0038±2e-6 | 0.0189±5e-5 |
| DiffWave | 0.0043±4e-4 | 0.0177±4e-4 |
| SSSD | **0.0034±4e-6** | 0.0119±1e-4 |
| DiffImp (Ours) | **0.0034±2e-5** | **0.0101±3e-4** |
| 20% RBM on PTB-XL | | |
| LAMC | 0.0759 | 0.1498 |
| CSDI | 0.0186±1e-5 | 0.0435±2e-4 |
| DiffWave | 0.0250±1e-3 | 0.0808±5e-3 |
| SSSD | 0.0103±3e-3 | 0.0226±9e-4 |
| DiffImp (Ours) | **0.0067±3e-5** | **0.0221±1e-3** |
| 20% BM on PTB-XL | | |
| LAMC | 0.0840 | 0.1171 |
| CSDI | 0.1054±4e-5 | 0.2254±7e-5 |
| DiffWave | 0.0451±7e-4 | 0.1378±5e-3 |
| SSSD | 0.0324±3e-3 | 0.0832±8e-3 |
| DiffImp (Ours) | **0.022±4e-5** | **0.059±1e-3** |

Table.15 shows the results of models trained with $C = 64$ at 300000 iterations and $C = 128$ at 150000 iterations. We can see that the two models achieve similar results on all four metrics of different missing ratios, which indicates a smaller $C$ leads to higher training cost.

Table 15: MSE, RMSE, MAE and MRE results of $C = 64$ (300000 iterations) and $C = 128$ (150000 iterations) on MuJoCo dataset with missing ratio 70%, 80% and 90%.

| | 90% | | | |
|---|---|---|---|---|
| | MSE | RMSE | MAE | MRE |
| $C = 64$ (300000iter) | 0.0008±0.00008 | 0.0277±0.0014 | 0.0126±0.0005 | 0.0121±0.0001 |
| $C = 128$ (150000iter) | 0.0004±0.00001 | 0.0191±0.0003 | 0.0142±0.0002 | 0.0146±0.0002 |
| | 80% | | | |
| | MSE | RMSE | MAE | MRE |
| $C = 64$ (300000iter) | 0.00030±0.00002 | 0.0174±0.0005 | 0.0104±0.0001 | 0.0107±0.0001 |
| $C = 128$ (150000iter) | 0.00031±0.00001 | 0.0178±0.0003 | 0.0114±0.0001 | 0.0117±0.0001 |
| | 70% | | | |
| | MSE | RMSE | MAE | MRE |
| $C = 64$ (300000iter) | 0.0003±0.00001 | 0.0166±0.0004 | 0.0117±0.0001 | 0.0121±0.0001 |
| $C = 128$ (150000iter) | 0.0004±0.00001 | 0.0191±0.0003 | 0.0142±0.0002 | 0.0146±0.0002 |

Fig.5 and Table.16 presents the inference time on ettm1 dataset with different sequence length. Fig.6 and Table.17 presents the inference time on MuJoCo and Electricity dataset with different number of channels. We can see from the result that the inference time is linear *w.r.t* the sequence length and number of channels, which demonstrates the linear complexity of our model. And Table.18 presents the results of 10%, 30% and 50% missing on MuJoCo dataset and 70%, 80% and 90% missing on Electricity dataset.

Table 16: Inference time of different sequence length on ettm1 dataset

| Sequence Length | Inference time (s) |
|---|---|
| 120 | 0.93 |
| 96 | 0.9252 |
| 480 | 0.944 |
| 576 | 0.946 |
| 1056 | 0.966 |

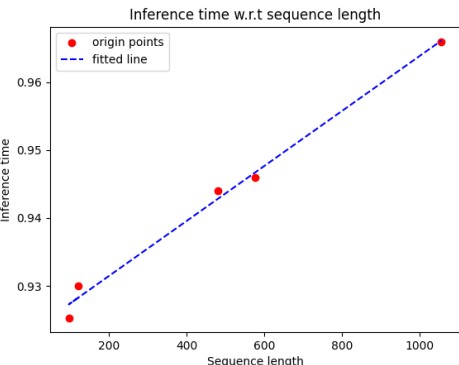

Figure 5: Inference time of different sequence length on ettm1 dataset

Table 17: Inference time of different number of channels on Mujoco and Electricity dataset

| MuJoCo | | Electricity | |
|---|---|---|---|
| Num of Channels | Inference time (s) | Num of Channels | Inference time (s) |
| 32 | 0.13 | 32 | 0.134 |
| 64 | 0.264 | 64 | 0.268 |
| 96 | 0.543 | 96 | 0.548 |
| 128 | 0.936 | 128 | 0.936 |

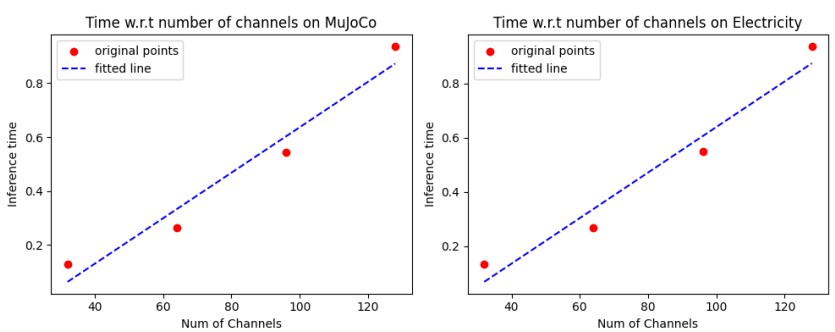

Figure 6: Inference time of different number of channels on Mujoco and Electricity dataset

Table 18: Experimental results of 10%, 30%, 50% on MuJoCo dataset and 70%, 80%, 90% on Electricity dataset

| | MuJoCo | | | Electricity | | |
|---|---|---|---|---|---|---|
| | 10% | 30% | 50% | 70% | 80% | 90% |
| MSE | 0.0003± 0.00001 | 0.0003±0.00001 | 0.0004±0.00001 | 1.469±0.0076 | 2.0085±0.0092 | 3.9443±0.0225 |
| RMSE | 0.0003± 0.0004 | 0.0182±0.00015 | 0.0187±0.0001 | 1.212±0.0031 | 1.4167±0.0030 | 1.9856±0.0058 |
| MAE | 0.0145±0.0002 | 0.0150±0.0001 | 0.0151±0.0001 | 0.7847±0.0013 | 0.9469±0.0023 | 1.3685±0.0033 |
| MRE | 0.0149±0.0002 | 0.0154±0.0001 | 0.0155±0.0001 | 0.4195±0.0006 | 0.5060±0.0002 | 0.7333±0.0007 |

### 7.2.3 HYPERPARAMETERS

Table 19 lists the hyperparameters in our model and ablation studies.

Table 19: Hyperparameters in DiffImp and ablation studies

| | DiffImp | DiffImp (In ablation studies) |
|---|---|---|
| Sequence dim ($C$ in Fig.2) | 128 | 64 |
| Residual channels ($K$ in Fig.2) | 128 | 64 |
| Num channels (dim of input projections before $\epsilon_\theta$) | 128 | 64 |
| Diffusion embedding dim | 128 | 128 |
| Training iteration | 150k | 450k |
| Num of conditional SMM | 1 | 1 |
| Num of input SMM | 1 | 1 |
| Num of sequential SMM | 1 | 1 |

### 7.3 EVALUATION METRIC DETAILS

In this part, we give details about the evaluation metrics in our experiments. As defined in Definition.1, the original time series is denoted as $y \in \mathbb{R}^{K \times L}$, the imputed time series is denoted as $\hat{y} \in \mathbb{R}^{K \times L}$, $M$ is the indicator matrix.

**Mean Absolute Error (MAE)**: MAE calculates the average $L_1$ distance between ground truth and the imputed values alongside the channel dimension, which is formulated as:

$$\mathbf{MAE}(y, \hat{y}) = \frac{1}{k} \sum_{i=1}^{K} \sum_{j=1}^{L} |(y - \hat{y}) \odot (1 - M)|_{i,j} \tag{21}$$

**Mean Square Error (MSE)**: MSE calculates the average $L_2$ between ground truth and the imputed values alongside the channel dimension, which is formulated as:

$$\mathbf{MSE}(y, \hat{y}) = \frac{1}{k} \sum_{i=1}^{K} \sum_{j=1}^{L} ((y - \hat{y}) \odot (1 - M))_{i,j}^2 \tag{22}$$

**Root Mean Square Error (RMSE)**: RMSE is the square root of RMSE:

$$\mathbf{RMSE}(y, \hat{y}) = \sqrt{\mathbf{MSE}(y, \hat{y})}$$
$$= \sqrt{\frac{1}{k} \sum_{i=1}^{K} \sum_{j=1}^{L} ((y - \hat{y}) \odot (1 - M))_{i,j}^2} \tag{23}$$

**Mean Relative Error (MRE)**: MRE estimates the relative difference between $y$ and $\hat{y}$:

$$\mathbf{MRE}(y, \hat{y}) = \frac{1}{k} \sum_{i=1}^{K} \sum_{j=1}^{L} (1 - M)_{i,j} \odot \frac{|(y - \hat{y})|_{i,j}}{y_{i,j}} \tag{24}$$

**Continuous Ranked Probabilistic Score (CRPS)**: Given an estimated probability distribution function $F$ modeled with an observation $x$, CRPS evaluates the compatibility and is defined as the integral of the quantile loss for all quantile levels:

$$\mathbf{CRPS}(F^{-1}, x) = \int_0^1 \Lambda_\alpha(F^{-1}(\alpha, x)) \, d\alpha, \tag{25}$$

where $\Lambda_\alpha(q, y) = (\alpha - \mathbf{1}_{y<q})(y - q), \alpha \in [0, 1]$ and $\mathbf{1}_{y<q}$ the indicator function, *i.e.*, if $y < q$, the value of the indicator function is 1, else 0.

Following Tashiro et al. (2021); Yan et al. (2024), we separate the interval $[0, 1]$ to 20 quantile levels with a stepsize of $s = 0.05$, and the estimated value of CRPS is:

$$\mathbf{CRPS}(F^{-1}, x) \approx \sum_{i=1}^{19} \frac{2\Lambda_{i \cdot s}(F^{-1}(i \cdot s, x))}{19} \tag{26}$$

For the whole time series $X \in \mathbb{R}^{K \times L}$, the CRPS value is normalized for all time steps and channels:

$$\mathbf{CRPS}(F^{-1}, X) = \frac{\sum_{i=1}^{K} \sum_{j=1}^{L} \mathbf{CRPS}(F_{i,j}^{-1}, X_{i,j})}{\sum_{i=1}^{K} \sum_{j=1}^{L} |X_{i,j}|} \tag{27}$$

**Continuous Ranked Probabilistic Score-Sum (CRPS-Sum)**: CRPS-sum calculates the CRPS for distribution $F$ for all $K$ features:

$$\mathbf{CRPS\text{-}Sum} = \frac{\sum_{j=1}^{L} \mathbf{CRPS}(F^{-1}, \sum_{i=1}^{k} X_{i,j})}{\sum_{i=1}^{K} \sum_{j=1}^{L} |X_{i,j}|} \tag{28}$$

## 7.4 ALGORITHM DETAILS OF BAM BLOCK AND CMB BLOCK

Alg.3 and Alg.4 describes the details of forward process in BAM and CMB block.

---

**Algorithm 3** Forward Process of BAM Block

---

1: **Input:** Time Representation Sequence $T_i \in \mathbb{R}^{B \times K \times L}$
2: **Output:** Time Representation Sequence $T_{i+1} \in \mathbb{R}^{B \times K \times L}$
3: {Normalize input sequence $T_i$}
4: $T_i' = \mathbf{Norm}(T_i)$
5: {Project $T_i'$ to target dim}
6: $x = \mathbf{Proj}_x(T_i')$
7: $w = \mathbf{Proj}_w(T_i')$
8: {Processing in different directions}
9: **For** $d$ in {forward,backward} **do**:
10:    **if** $d = $ forward:
11:       $T_f = \mathbf{Mamba}(T_i')$
12:    **if** $d = $ backward:
13:       $T_d = \mathbf{Flip}(T_i')$
14:       $T_d = \mathbf{Mamba}(T_d)$
15: **End For**
16: {Learning weights for different positions}
17: $w = \mathbf{Proj}_a(w)$
18: $w = \mathbf{Sigmoid}(w)$
19: {Temporal attention}
20: $T_d = w \odot T_d$
21: $T_f = w \odot T_f$
22: {Feature fusion}
23: $T_o = T_d + T_f$
24: $T_o = \mathbf{Proj}_o(T_o)$
25: {Residual connection}
26: $T_{i+1} = T_o + T_i$

---

## 7.5 VISUALIZATION RESULTS

---

**Algorithm 4** Forward Process of CMB Block

---

1: **Input:** Time Representation Sequence $T_i \in \mathbb{R}^{B \times K \times L}$
2: **Output:** Time Representation Sequence $T_{i+1} \in \mathbb{R}^{B \times K \times L}$
3: {Transpose the channel dimension: $\mathbb{R}^{B \times K \times L} \to \mathbb{R}^{B \times L \times K}$}
4: $T_i = \mathbf{Transpose}(T_i)$
5: {Normalize input sequence $T_i$}
6: $T_i' = \mathbf{Norm}(T_i)$
7: {Project $T_i'$ to target dim}
8: $x = \mathbf{Proj}_x(T_i')$
9: $w = \mathbf{Proj}_w(T_i')$
10: $T_f = \mathbf{Mamba}(T_i')$
11: {Learning weights for different positions}
12: $w = \mathbf{Proj}_a(w)$
13: $w = \mathbf{Sigmoid}(w)$
14: {Temporal attention}
15: $T_f = w \odot T_f$
16: $T_o = \mathbf{Proj}_o(T_f)$
17: {Residual connection}
18: $T_{i+1} = T_o + T_i$
19: {Transpose the channel dimension: $\mathbb{R}^{B \times L \times K} \to \mathbb{R}^{B \times K \times L}$}
20: $T_{i+1} = \mathbf{Transpose}(T_i)$

---

Figure 7: Visualization of probabilistic imputation results on MuJoCo dataset across all 14 channels with missing ratio 90%

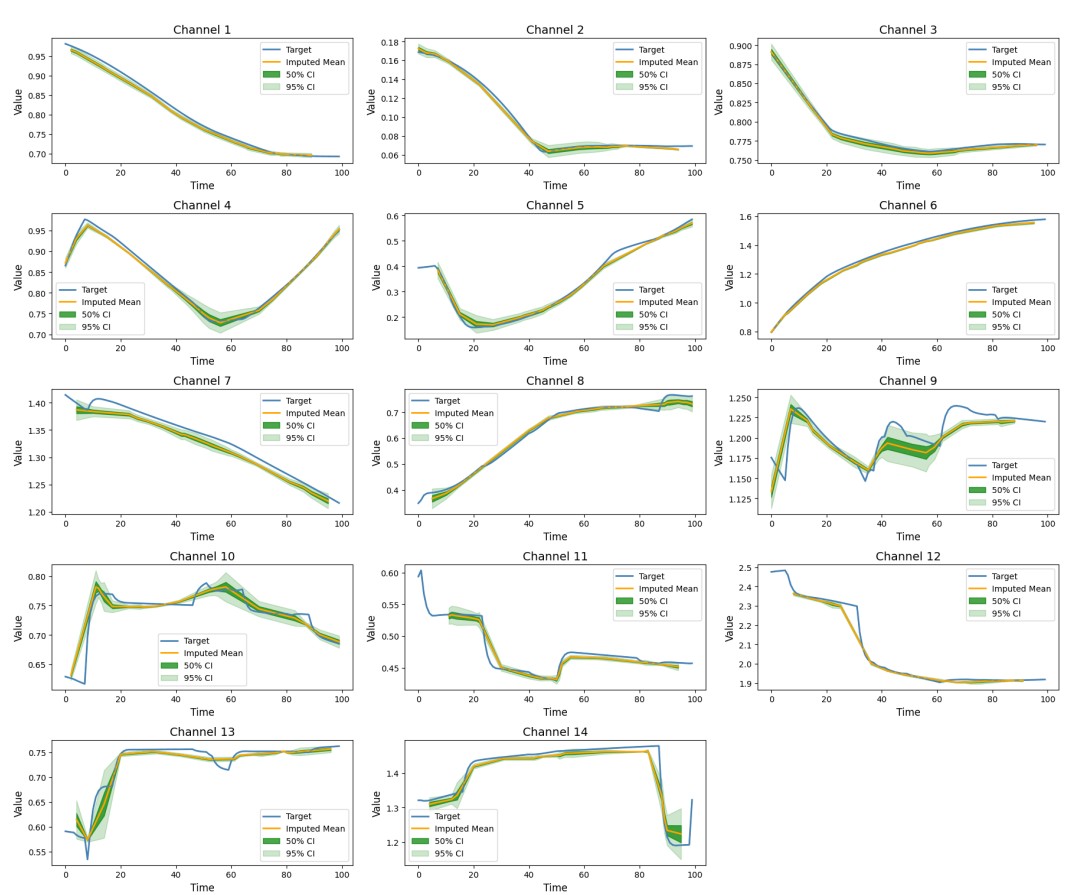

Figure 8: Visualization of probabilistic imputation results on MuJoCo dataset across all 14 channels with missing ratio 80%

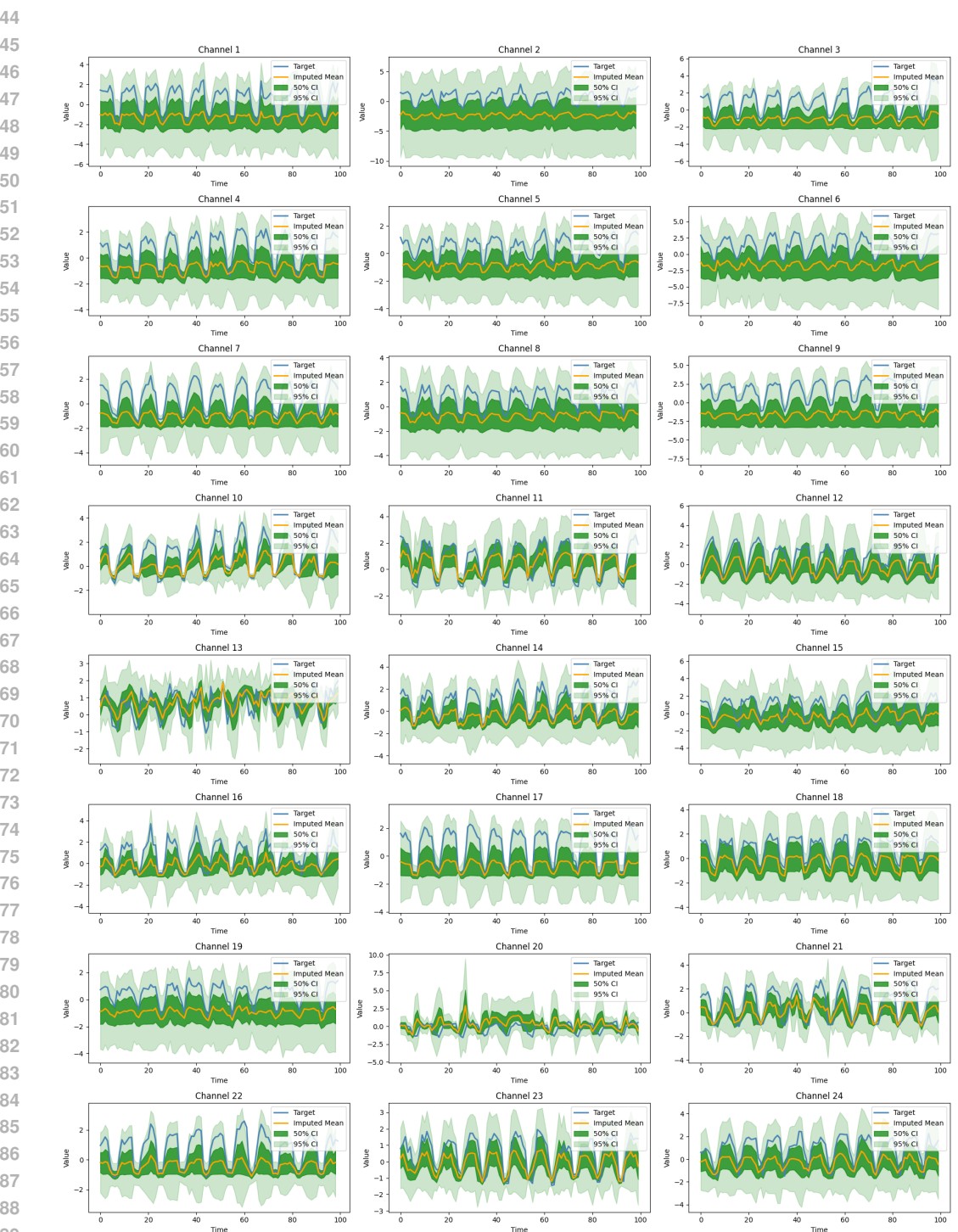

Figure 9: Visualization of probabilistic imputation results on Electricity dataset across the first 24 channels with missing ratio 10%

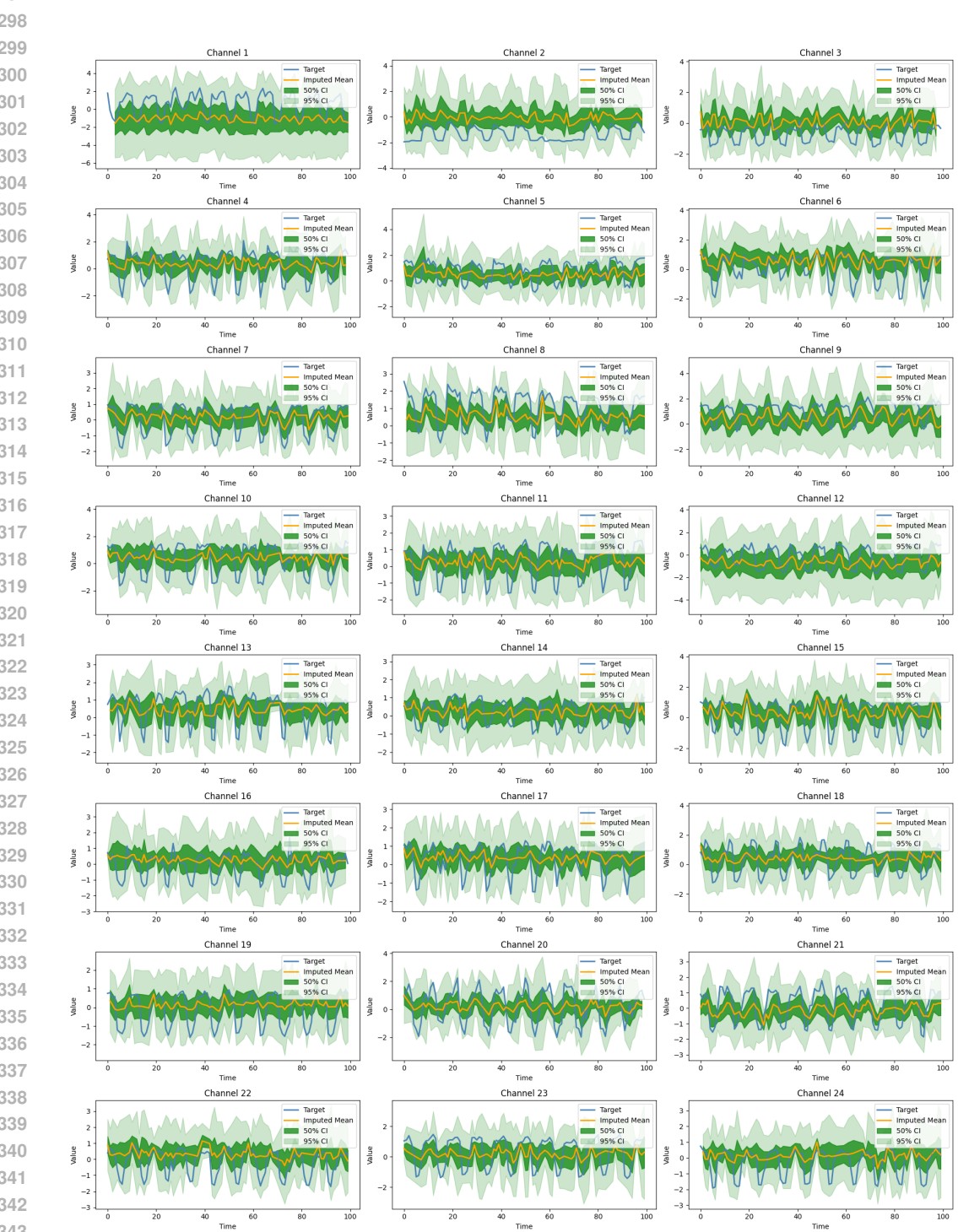

Figure 10: Visualization of probabilistic imputation results on Electricity dataset across the first 24 channels with missing ratio 30%

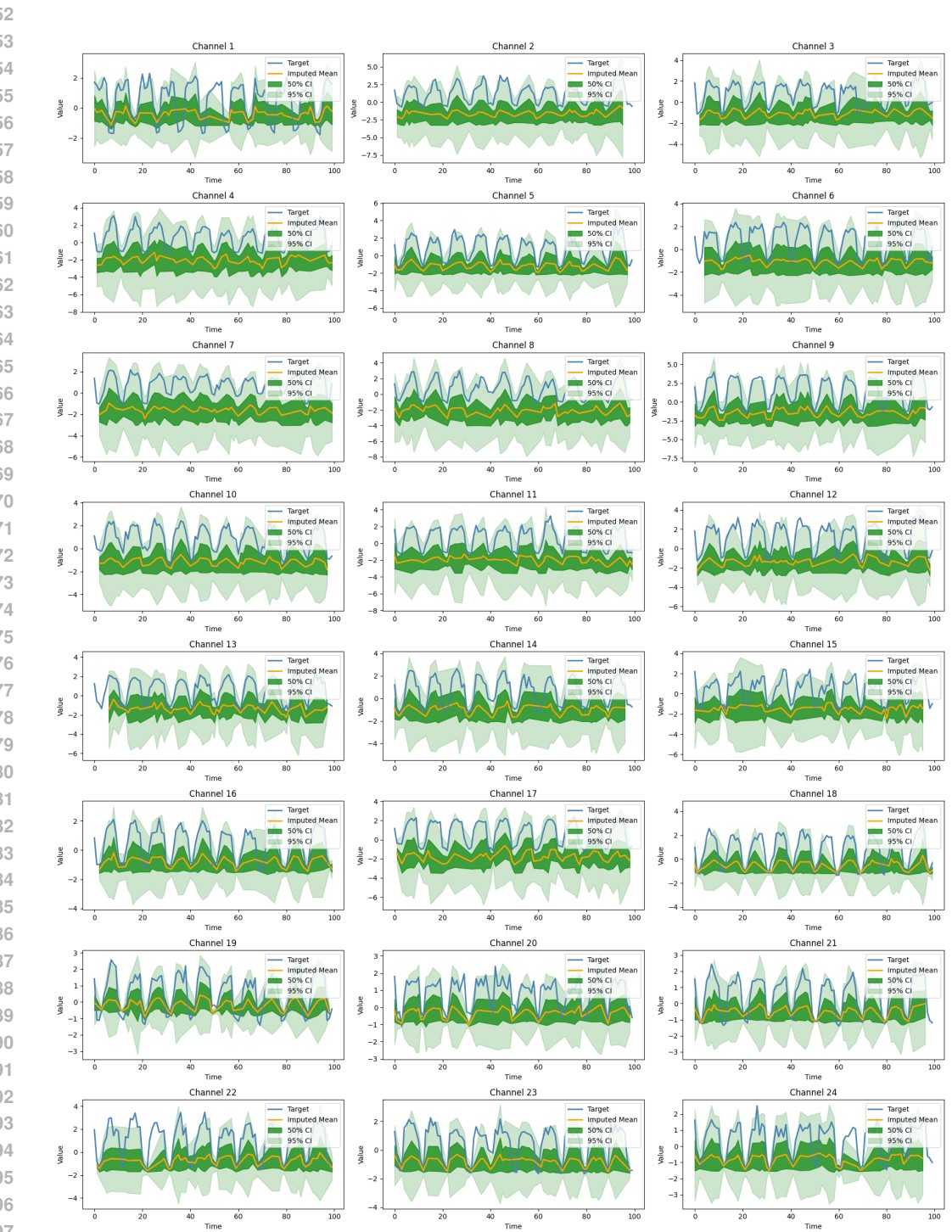

Figure 11: Visualization of probabilistic imputation results on Electricity dataset across the first 24 channels with missing ratio 50%

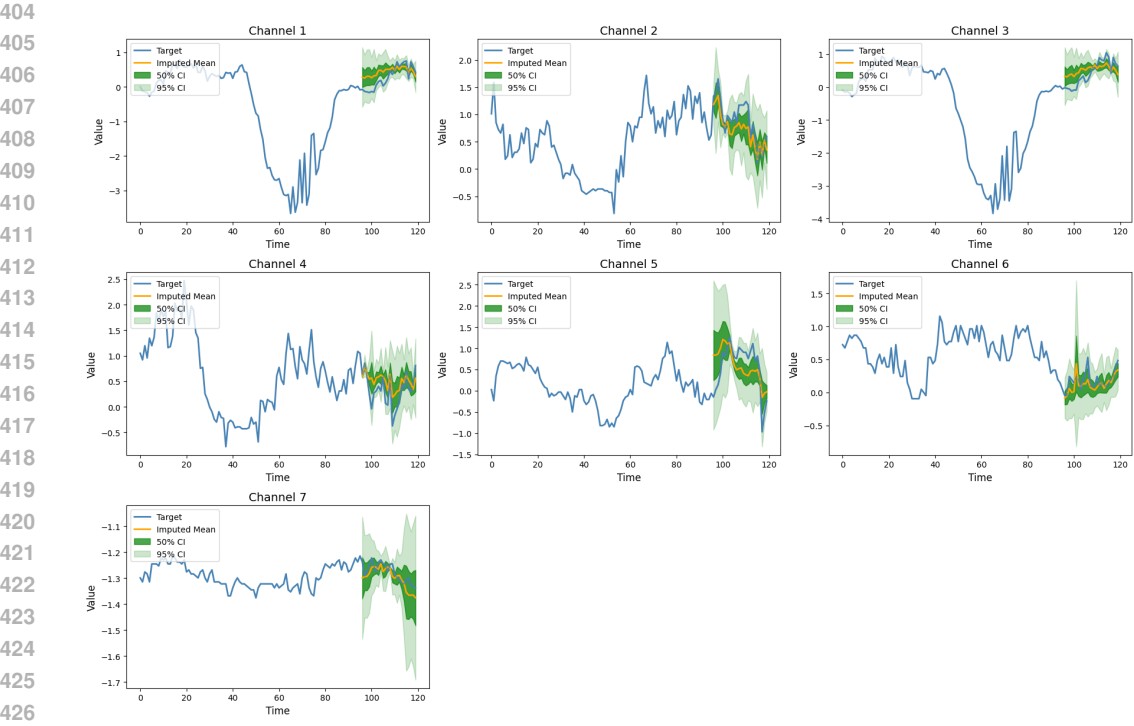

Figure 12: Visualization of probabilistic forecasting results on ETTm1 dataset across all 7 channels with forecasting length 24

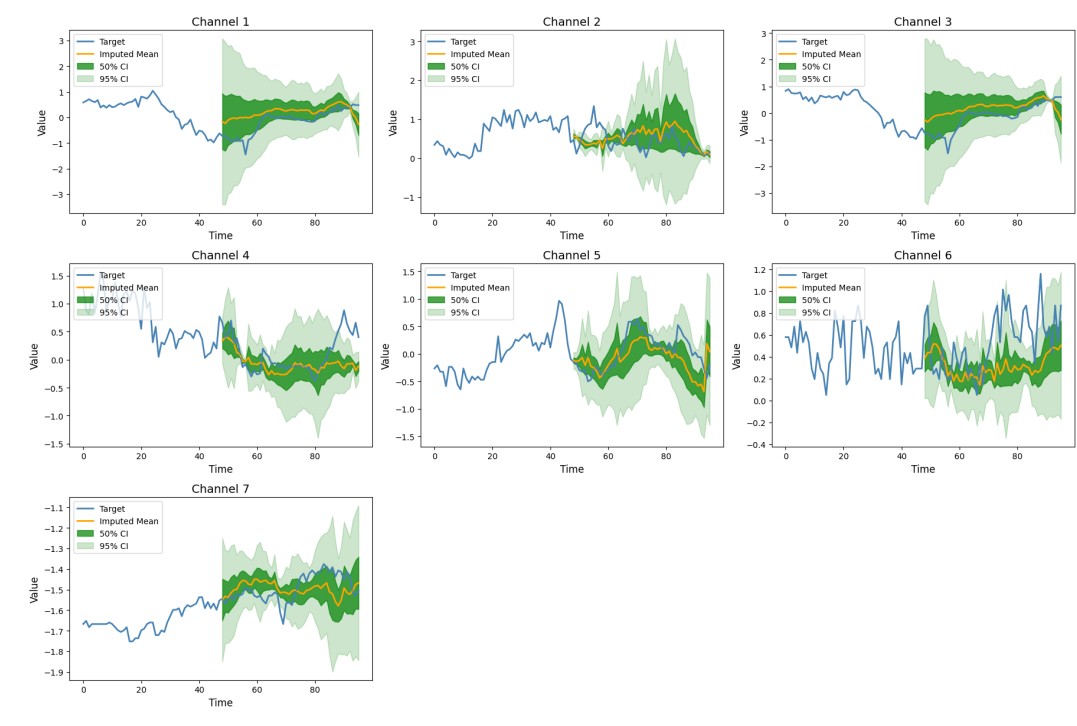

Figure 13: Visualization of probabilistic forecasting results on ETTm1 dataset across all 7 channels with forecasting length 48

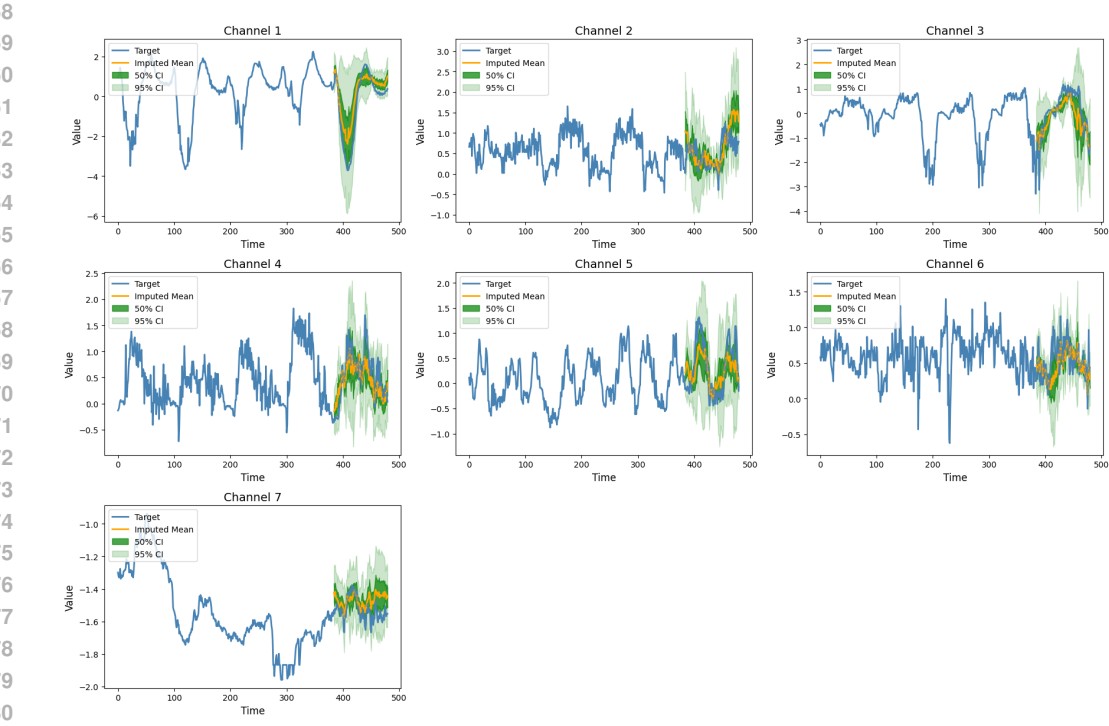

Figure 14: Visualization of probabilistic forecasting results on ETTm1 dataset across all 7 channels with forecasting length 96

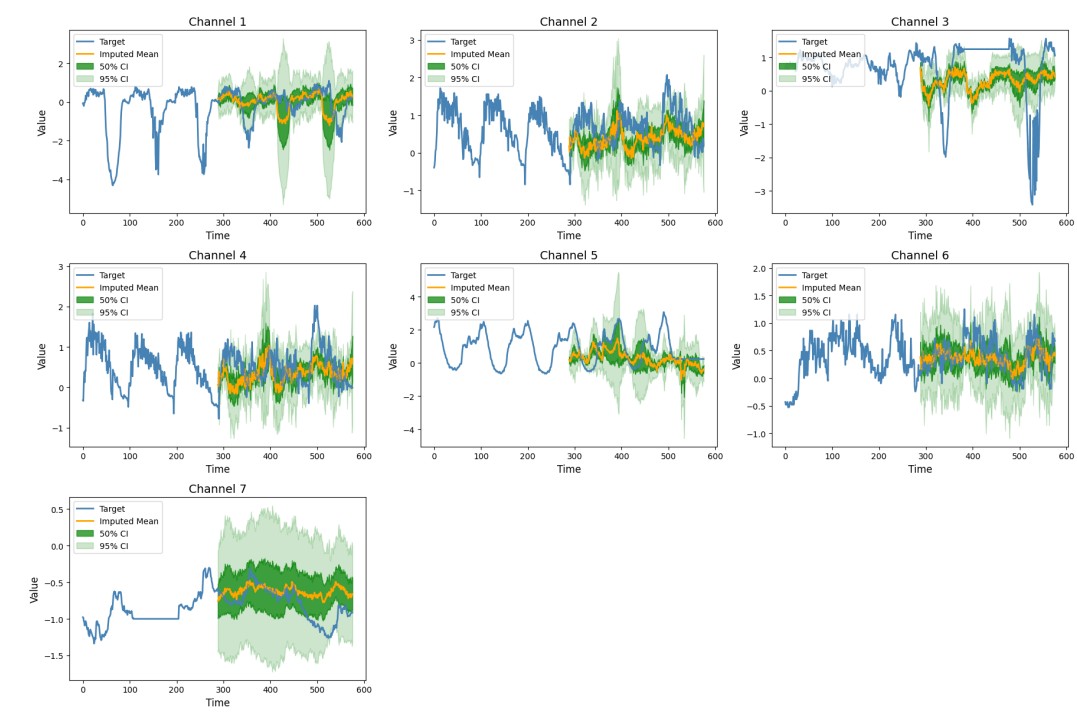

Figure 15: Visualization of probabilistic forecasting results on ETTm1 dataset across all 7 channels with forecasting length 288

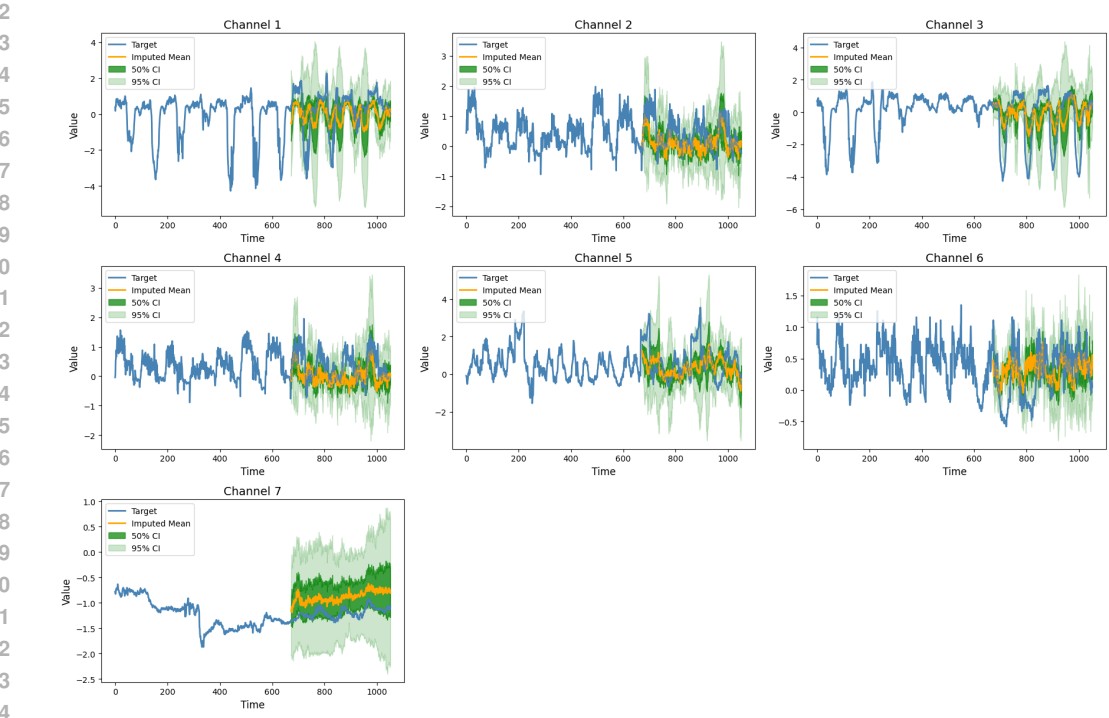

Figure 16: Visualization of probabilistic forecasting results on ETTm1 dataset across all 7 channels with forecasting length 672

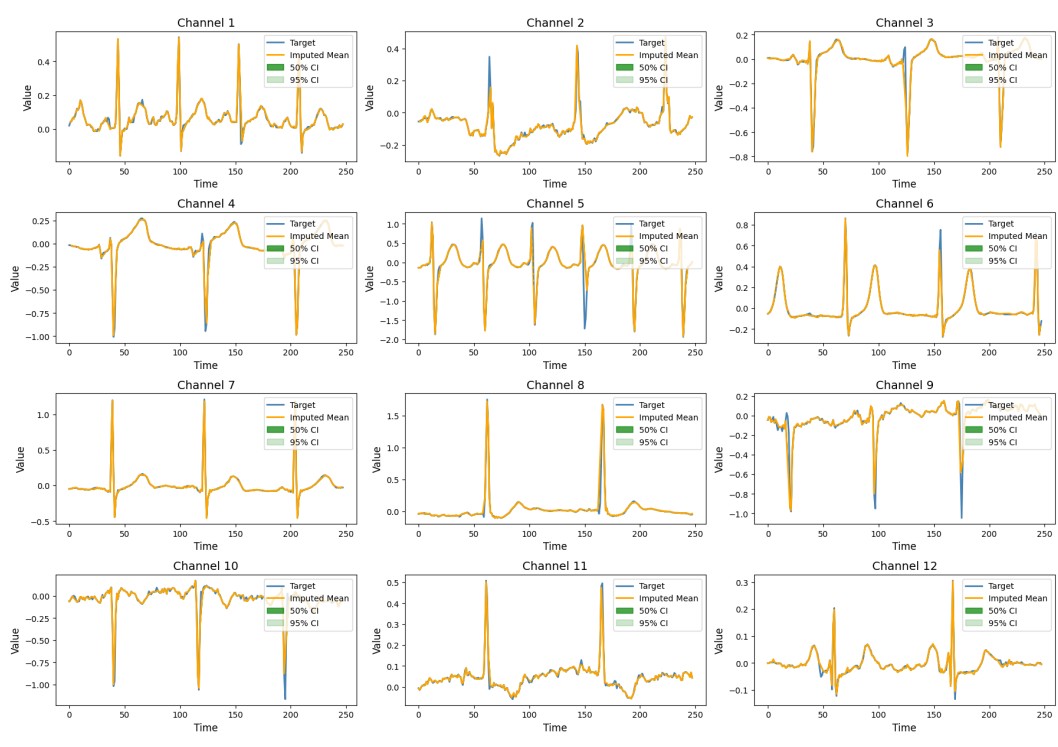

Figure 17: Visualization of probabilistic imputation results on PTB-XL dataset across all 12 channels with missing ratio 20%

