# OpenReview forum: "DiffImp: Efficient Diffusion Model for Probabilistic Time Series Imputation with Bidirectional Mamba Backbone"
_ICLR.cc/2025/Conference — Submitted to ICLR 2025_

### Official Review · Reviewer_ZcoV · 2024-10-28

**Soundness:** 3
**Presentation:** 3
**Contribution:** 3
**Rating:** 8
**Confidence:** 4

**Summary:**

This paper provides an efficient method for time series imputation problems with a self-supervised training manner. The authors implemented the state-space model based Mamba architecture as the backbone for diffusion models. The authors validate their model on different real-world datasets.

**Strengths:**

1. This paper proposes an effective method for the time series data and the architecture details (bidirectional modeling and channel dependency) is well-suited for the time series data.
2. For probablistic time series imputation, the proposed method has a significant improvement in the CRPS-sum metric.

**Weaknesses:**

1. The implementation details of probabilistic time series imputation is not clearly explained in the paper.

2. The diffimp seems like a condition diffusion model but the conditions are not clearly explained in the paper.

3. While modeling inter-channel dependencies, it is clear that the dependency is related to the distance of the timestep, is diffimp capable of addressing the distance issue?

**Questions:**

1. You have mentioned your model is a probabilistic model but the analysis of probabilistic imputation is not clear. Can the authors provide more details about why your model is capable of achieving probabilistic impuation and the implementation of the probabilistic time series imputation?

2. Can the author provide more details about the choice of mask ratios among the datasets and how the inference time is calculated among different methods?

3. The model is a conditional dpm. Can the authors provide more details about the conditions and how the conditions are used in the model?

4. A straightforward conclusion is that different distance has different impacts on one timestamp. Can the authors provide more details about the inter-channel dependencies and how the model addresses the distance issue?

---

> ### Author Response · Authors · 2024-11-24
> **Response to Reviewer ZcoV (Part I)**
>
> We appreciate the time and effort Reviewer ZcoV dedicated to evaluating our manuscript and we are thankful to Reviewer ZcoV for their thoughtful comments on the implementation details, experimental details about our paper. We have revised our paper accordingly.
>
> **Q1:** You have mentioned your model is a probabilistic model but the analysis of probabilistic imputation is not clear. Can the authors provide more details about why your model is capable of achieving probabilistic impuation and the implementation of the probabilistic time series imputation?
>
> **A1:** Thank you for your comments on the probabilistic time series imputation details. Here is the explaination of why our model is capable of achieving probabilistic time series imputation.
>
> The key point of probabilistic time series imputation is to accurately model the distribution of the unknown points. Our method is based on a diffusion model, which has a strong ability to model distributions. In our model, we use the diffusion model to model the posterior distribution of the missing points relative to the observed points. By sampling from the modeled posterior distribution, the mean and standard deviation of multiple samples can be regarded as the mean and standard deviation of the missing points, thus achieving probabilistic time series imputation.
>
> **Q2:** Can the author provide more details about the choice of mask ratios among the datasets and how the inference time is calculated among different methods?
>
> **A2:** Thank you for your feedback on experimental details about our paper. 2. To maintain consistency with the results of the imputation method using state-space models, we select the datasets used in [1] and retain the baseline consistent with [1]. We conduct uncertainty imputation tests on the Mujoco dataset (with 70%, 80%, and 90% missing rates) and the Electricity dataset (with 10%, 30%, and 50% missing rates), and evaluate the CRPS-sum metric on the Electricity dataset. Additionally, to remain consistent with the imputation baseline, we perform supplementary experiments on three datasets: PTB-XL (ECG data), air quality, and healthcare. The experimental results are shown in the table below.
> As for sampling time, to ensure consistency with other baselines, we evaluate the model by running it multiple times ( set as 5 in our experiment) with a large batch size (batchsize = 100) and multiple samples (nsample) in a batch and calculate the average time **T**. As shown in  **Table 8 in the revised paper** , the time reported in our paper refers to the time required to generate a single sample within one batch, i.e.,$\frac{T}{batchsize \times nsample}$
>
> [1] Juan Miguel Lopez Alcaraz and Nils Strodthoff. Diffusion-based time series imputation and forecasting with structured state space models.

---

> ### Author Response · Authors · 2024-11-24
> **Response to Reviewer ZcoV (Part II)**
>
> **Q3:** The model is a conditional dpm. Can the authors provide more details about the conditions and how the conditions are used in the model?
>
> **A3:** Thank you for your comments on model details of our paper. **Fig. 1** illustrates how our model constructs conditional inputs using input data. For the original data, if it does not contain any missing value, we first apply a masking operation to part of the original data, effectively splitting it into training and test sets. For the training set, we further divide it with the same proportion. The missing portion (the masked part) is the target for imputation during the training process, while the unmasked part of the training set serves as the model's conditional input. On the other hand, after obtaining the conditional input, the conditional input, step embedding, and noisy input together form parameterized noise to update the model parameters.
>
> **Q4:** A straightforward conclusion is that different distance has different impacts on one timestamp. Can the authors provide more details about the inter-channel dependencies and how the model addresses the distance issue?
>
> **A4:** Thank you for your insightful comments on our paper. The distance issue and inter-channel dependencies can be explained as follows: for information interaction within the channel, the objective is to enable the model to fully leverage global information, including data from both past and future time steps relative to the missing values. To achieve this, we employ a bidirectional SSM-based model as the backbone of diffusion models. The forward part records all information from past time steps relative to the missing value, while the backward part captures information from future time steps, thereby enabling bidirectional information modeling. Regarding the relative positional relationships and the distance issue, we analyze it from two perspectives. First, the SSM-based Mamba model inherently models relative positional relationships (i.e. the distance relationship) within its architecture. Second, we design a temporal attention module that generates position-specific weights based on the input, further incorporating the relative positional relationships.

---

> > ### Comment · Reviewer_ZcoV · 2024-11-24
> >
> > The authors' responses effectively address my previous concerns regarding the methodology and details. The additional results on the extra datasets also demonstrate the effectiveness and efficiency of the proposed method.  Therefore, I am pleased to adjust my rating accordingly.

---

### Official Review · Reviewer_Aeb7 · 2024-10-31

**Soundness:** 3
**Presentation:** 3
**Contribution:** 3
**Rating:** 8
**Confidence:** 4

**Summary:**

This paper provides a diffusion-based imputation method for time series data with missing values. The proposed method integrates Mamba-based models and diffusion-based models to achieve linear complexity for sequence data. The method is evaluated on two real-world datasets, Electricity and MuJoco, and the results show that it outperforms other state-of-the-art methods in terms of accuracy and efficiency.

**Strengths:**

1. This paper proposes an effective method for the time series data. The authors explores the availabity of using Mamba models as the diffusion backbones. The intergration of Mamba-based models and diffusion-based models provides linear complexity for sequence data and works fine on time series imputation tasks.
2. The improvements of deterministic results on high missing ratios are significant and the CRPS-sum improvement is also very significant compared with the other baselines, which means the proprosed method is effective in .
3. The running time of the proposed method is significantly lower than the other baselines and demonstrates the efficiency of the proposed method.
4. The bidirectional modeling, temporal attention module and channel modeling module are well designed and can effectively capture the temporal and channel dependencies of the time series data.
5. The authors provide detailed code for reproducible experiments.

**Weaknesses:**

1. In your paper, the experiments are conducted on the Electricity and MuJoco dataset. And as in [1], the experiments are conducted on AQI and healthcare dataset. The reason of choosing these datasets seems not clear.

2. for efficiency, the authors only compare the time complexity of the proposed method with the state-of-the-art methods.

3. The training manner of building missing values may need further expalaination

[1] CSDI: Conditional Score-based Diffusion Models for Probabilistic Time Series Imputation, Tashiro et al(2021)

**Questions:**

For contents:

1. the authors state the mask strategy is utilized for training and in all there are three kinds of mask strategies. It is not clear that how you applied the mask. e.g. if there are no missing values in the original dataset, how to construct the mask for training? and if there are missing values in the original dataset, how to construct the mask for training? Now it seems that the mask is applied similarly to all kinds of datasets.

2. It seems that you designed two kinds of blocks for modeling channel dependency and sequential dependency, why you choose unidirectional channel modelling as shown in Fig 3c.

3. As shown in Fig 3d, it seems not clear how the temporal attention is utilized for the bidirectional modelling, besides, as in Fig 3c, the CMB module has the same sigmoid module as Fig 3d, is it also the temporal attention module and why you applied temporal attention module for channel modeling?

For experiments

1. Can the authors explain the choice of datasets and provide further experiments results and comparsions on AQI and healthcare dataset?

2. As it is an efficient time series diffusion model, can the authors provide more details, including training memory details or model parameters?

For details

4. as in table 1, the authors state that the SSM models is partial global dependency, why?

---

> ### Author Response · Authors · 2024-11-24
> **Response to Reviewer Aeb7 (Part I)**
>
> We sincerely appreciate Reviewer Aeb7 for the constructive suggestions and careful evaluation of our work and we are grateful for the detailed comments on experiments, contents and details. We have revised our paper accordingly.
>
> **For contents:**
>
> **Q1:**  the authors state the mask strategy is utilized for training and in all there are three kinds of mask strategies. It is not clear that how you applied the mask. e.g. if there are no missing values in the original dataset, how to construct the mask for training? and if there are missing values in the original dataset, how to construct the mask for training? Now it seems that the mask is applied similarly to all kinds of datasets.
>
> **A1:** Thank you for your insight comments on the mask strategy. Consistent with [1], in our experiments, we use three different masking strategies to create the training datasets: random missing, random block missing, and blackout missing (in  **Sec 3.1**). For complete datasets, we first randomly partition a portion of the data according to the predefined masking strategy as the test set, and use the remaining portion as the training set. Within the training set, we further partition the data using the same strategy and ratio. During training, the training set is treated as a complete dataset, and the masked parts are imputed. After training, we test it on the partitioned test set. For incomplete datasets (i.e., datasets that already contain missing values), we do not partition the data into training and test sets; instead, we apply the same missing rate and masking strategy to the training set during training to achieve self-supervised learning.
>
> [1] Juan Miguel Lopez Alcaraz and Nils Strodthoff. Diffusion-based time series imputation and forecasting with structured state space models.
>
> **Q2:** It seems that you designed two kinds of blocks for modeling channel dependency and sequential dependency, why you choose unidirectional channel modelling as shown in Fig 3c.
>
> **A2:** Thank you for your feedback on the modeling manner in our model. The reason we choose unidirectional modeling is as follows: In our module, we employ two different approaches for modeling inter-channel and intra-channel dependencies. Based on the channel dependencies inherent in time series and an analysis of the position of the missing points in the imputation problem, we observe that known points may appear on both sides of the missing points. Therefore, to model the bidirectional dependencies, we use a bidirectional module for sequence properties. However, for the inter-channel dependencies, the relative position does not directly correlate with the dependency structure. Hence, for channel relationships, we adopt a unidirectional modeling approach.
>
> **Q3:** As shown in Fig 3d, it seems not clear how the temporal attention is utilized for the bidirectional modelling, besides, as in Fig 3c, the CMB module has the same sigmoid module as Fig 3d, is it also the temporal attention module and why you applied temporal attention module for channel modeling?
>
> **A3:** Thank you for your insightful comments on the attention modules. As shown in  **Fig.3d** , the temporal attention module within the dashed box can be described as follows: the network passes the input through a linear projection module, and the output of the projection is passed through an activation function (sigmoid in experiments). This results in a value within the range of $[-1, 1]$, with the same size as the input. We treat this value as a weight to be assigned to different positions in the input, thereby implementing temporal attention. In  **Fig.3c** , the forward propagation of the module is the same as in  **Fig.3d** , but since its input processes channel data (by transposing the input), the module in **Fig.3c** implements a channel attention to adjust the weight relationships between channels.

---

> ### Author Response · Authors · 2024-11-24
> **Response to Reviewer Aeb7 (Part II)**
>
> **For Experiments:**
>
> **Q1:** Can the authors explain the choice of datasets and provide further experiments results and comparsions on AQI and healthcare dataset?
>
> **A1:** Thank you for your insightful comments on baselines and datasets. We agree with the reviewer that we should also include some basic baselines and datasets as proposed in the comment, which may demonstrate the effectiveness of our methods and make our contributions more convincing.  We have made the following revisions according to your comment:
>
> 1. We add experiements on PhysioNet dataset and air quality dataset in **Table 11,12,13 in the revised paper**.
> 2. We add comparisons with more time series imputation methods in **Table 11 in the revised paper**.
>
> This table shows the MAE and MSE results on Air quality dataset, including more time series imputation baselines. It can be observed that on the Air quality dataset, our method achieves the best MAE performance, with a performance improvement of 30% compared to the second best method. As for the MSE metric, out method get the second best result and is only 1.85% higher than the best result.
>
> |         | AQI       |AQI    |
> |---------|----------------|-------------------|
> |         | MAE    | MSE   |
> | V-RIN   | 25.4 $\pm$ 0.62  | -    |
> | GP-VAE  | 25.71    | 2589.53   |
> | BRITS   | 14.1$\pm$ 0.26  | 495.94$\pm$ 43.56  |
> | SPIN    | 11.77$\pm$ 0.54 | -      |
> | SPIN-H  | 10.89$\pm$ 0.27 | -   |
> | gatgpt  | 10.28     | **341.26**    |
> | GRIN    | 10.51$\pm$ 0.28 | 371.47$\pm$ 17.38  |
> | CSDI    | 9.60$\pm$ 0.04  | -      |
> | DiffImp | **6.75$\pm$ 0.014** | 347.58$\pm$ 0.55    |
>
>  '-' denotes the corresponding result is not provided in the original paper.
>
> The following table shows the RMSE results on Air quality and Physionet (healthcare) dataset. On the Physionet dataset, our method achieves SOTA RMSE performance on all three missing ratios, with improvement of 22.6%, 17.2%, 23.5% correspondingly. On the Air quality dataset, our method achieves the second best RMSE result. The MSE, MAE and RMSE results demonstrate the effectiveness of our models on time series imputation problems.
>
> |         | Physionet       |   Physionet        |     Physionet  | AQI    |
> |---------|-------------------|-------------------|-------------------|-----------------|
> |         | 10% missing       | 50% missing       | 90% missing       |                 |
> | V-RIN   | 0.628$\pm$ 0.025  | 0.693$\pm$ 0.022  | 0.928$\pm$ 0.013  | 40.11$\pm$ 1.14 |
> | BRITS   | 0.619$\pm$  0.018  |  0.701$\pm$ 0.021 | 0.847$\pm$ 0.021  | 24.28$\pm$ 0.65 |
> | SSGAN   | 0.607$\pm$ 0.034  | 0.758 $\pm$ 0.025 | 0.830$\pm$ 0.009  | -               |
> | RDIS    | 0.635$\pm$ 0.018  | 0.747 $\pm$ 0.013 | 0.922$\pm$ 0.018  | 37.25$\pm$ 0.31 |
> | CSDI    | 0.531$\pm$ 0.009  | 0.668$\pm$ 0.007  | 0.834$\pm$ 0.006  | 19.21$\pm$ 0.13 |
> | CSBI    | 0.547$\pm$ 0.019  | 0.649 $\pm$ 0.009 | 0.837$\pm$ 0.012  | 19.07$\pm$ 0.18 |
> | SSSD    | 0.459$\pm$ 0.001  | 0.632$\pm$ 0.004  | 0.824$\pm$ 0.003  | 18.77$\pm$ 0.08 |
> | TS-Diff | 0.523$\pm$ 0.015  | 0.679$\pm$ 0.009  | 0.845$\pm$ 0.007  | 19.06$\pm$ 0.14 |
> | SAITS   | 0.461$\pm$ 0.009  | 0.636$\pm$ 0.005  | 0.819$\pm$ 0.002  | 18.68$\pm$ 0.13 |
> | D^3M    | 0.438$\pm$ 0.003  | 0.615$\pm$ 0.012  | 0.814$\pm$ 0.002  | **18.19$\pm$ 0.18** |
> | TIDER   | 0.486$\pm$ 0.006  | 0.659$\pm$ 0.009  | 0.833$\pm$ 0.005  | 18.94$\pm$ 0.21 |
> | DiffImp | **0.339$\pm$ 0.0002** | **0.509$\pm$ 0.007**  | **0.623$\pm$ 0.0001** | 18.66$\pm$ 0.26 |
>
> The following table exhibits the CRPS results on Air quality and Physionet (healthcare) dataset. Our method performs SOTA on 10%, 50% missing of Physionet dataset and AQI dataset. The CRPS results indicates that our method has strong capability of handling probabilistic time series imputation problem.
>
> |         | Physionet         |     Physionet     |    Physionet           | AQI             |
> |---------|-------------------|---------------------|-------------------|--------------------|
> |         | 10% missing       | 50% missing         | 90% missing       |                    |
> | GP-VAE  | 0.582$\pm$ 0.003       | 0.796$\pm$ 0.004         | 0.998$\pm$ 0.001       | 0.402$\pm$ 0.009        |
> | V-RIN   | 0.814$\pm$ 0.004       |  0.845$\pm$ 0.002        | 0.932$\pm$ 0.001       | 0.534$\pm$ 0.013        |
> | CSDI    | 0.242$\pm$ 0.001       | 0.336$\pm$ 0.002         | 0.528$\pm$ 0.003       | 0.108$\pm$ 0.001        |
> | CSBI    | 0.247$\pm$ 0.003       | 0.332 $\pm$ 0.003        | 0.527$\pm$ 0.006       | 0.110$\pm$ 0.002        |
> | SSSD    | 0.233$\pm$ 0.001       | 0.331$\pm$ 0.002         | 0.522$\pm$ 0.002       | 0.107$\pm$ 0.001        |
> | TS-Diff | 0.249$\pm$ 0.002       | 0.348$\pm$ 0.004         | 0.541$\pm$ 0.006       | 0.118$\pm$ 0.003        |
> | D^3M    | 0.223$\pm$ 0.001       | 0.327$\pm$ 0.003         | **0.520$\pm$ 0.001**       | 0.106$\pm$ 0.002        |
> | DiffImp | **0.164$\pm$ 0.0004** | **0.2438$\pm$ 0.00008** | 0.533$\pm$ 0.0004 | **0.0959$\pm$ 0.0002** |

---

> ### Author Response · Authors · 2024-11-24
> **Response to Reviewer Aeb7 (Part III)**
>
> **Q2:** As it is an efficient time series diffusion model, can the authors provide more details, including training memory details or model parameters?
>
> **A2:** Thank you for your comments on efficiency. We have update the training memory and model parameter details in **Table 8 in the revised paper** and we also list it here:
>
> We can see from the table that CSDI has the lowest inference time because it contains the fewest parameter.
> Comparing with SSSD (diffusion-based time series imputation model), the inference time and memory cost of our model is almost the same if we modify our models to make them contains nearly same number of parameters. Secondly, we can see that inference time and gpu memory cost is linear with respect to the number of Channels.
>
> |                      |     CSDI    |   CSDI     |     SSSD    | SSSD   | DiffImp (C=64) | DiffImp (C=64)  | DiffImp (C=96) | DiffImp (C=96)  | DiffImp (C=128) | DiffImp (C=128) |
> |:--------------------:|:-----------:|:------:|:-----------:|:------:|:--------------:|:------:|:--------------:|:------:|:---------------:|:------:|
> |                      | electricity | MuJoCo | electricity | MuJoCo |   electricity  | MuJoCo |   electricity  | MuJoCo |   electricity   | MuJoCo |
> |    Model size (M)    |     0.05    |  0.05  |    49.23    |  48.3  |      24.21     |   24   |      51.03     |  50.92 |       87.7      |  87.57 |
> |   Inference time(s)  |     0.10    |  0.051 |     0.42    |  0.416 |      0.268     |  0.264 |      0.548     |  0.543 |      0.936      |  0.936 |
> | GPU Memory Cost (MB) |   4046     |  3226  |     2534    |  2448  |      1662      |  1748  |      2724      |  2696  |       4604      |  4574  |
>
> **For details:**
>
> **Q1:** as in table 1, the authors state that the SSM models is partial global dependency, why?
>
> **A1:** Thank you for your comments on our model details. In the forward process of SSM, as shown in **Eq.1** and  **Eq.2** , we observe that the model utilizes all the information from past time steps, which indicates that the model has the ability to capture global information. Therefore, the SSM model has global dependency. On the other hand, for time step $t$ , the model can only access information from the start time up to time $t$, and cannot access information from future time steps. Thus, we refer to this as partial global dependency.

---

> > ### Comment · Reviewer_Aeb7 · 2024-11-24
> > **Thanks for rebuttal**
> >
> > I would like to thank the author for their detailed rebuttal. My concerns have been well addressed. Then I will keep my score at 8, a good paper for recommendation acceptance. Thanks.

---

### Official Review · Reviewer_tnPK · 2024-11-01

**Soundness:** 2
**Presentation:** 2
**Contribution:** 1
**Rating:** 3
**Confidence:** 4

**Summary:**

Probabilistic time series imputation, capable of estimating imputation uncertainty, is widely used in real-world applications. Denoising diffusion probabilistic models excel in this task due to their ability to model complex distributions. However, current DDPM-based methods face challenges in achieving low time complexity for sequence modeling and effectively handling inter-variable and bidirectional dependencies. To overcome these, the authors integrate the efficient state space model Mamba as the backbone denoising module and design SSM-based blocks for bidirectional modeling and inter-variable relation understanding. Experimental results show that the approach achieves state-of-the-art time series imputation results across multiple datasets, missing scenarios, and missing ratios.

**Strengths:**

S1. Time series imputation is an important and widely studied problem.

S2. Diffusion model and Mamba are popular models recently.

S3. Complexity analysis is provided for the model.

**Weaknesses:**

W1. As discussed in Section 1, the main contribution is proposing a diffusion-based framework for time series imputation, by integrating Mamba-based blocks into diffusion models. However, diffusion models have been proved to be effective in time series imputation, and Mamba is also an existing technique. Modeling bidirectional dependencies and channel dependencies are common techniques to capture characteristics of time series data. Therefore, the novelty and contributions are limited.

W2. There is no theoretical result to support the proposed techniques.

W3. It is unbelievable that the time cost is totally ignored in experiment results, since the main contribution claimed by authors is that the linear complexity of proposed models.

W4. Only synthetic missing values are considered in experiments. It is suggested to use real-world incomplete datasets to demonstrate the effectiveness and efficiency of proposed methods.

W5. 70%, 80%, 90% missing values are injected into the MuJoCo dataset, and 10%, 30%, 50% missing values are considered into the Electricity dataset. It is wondered why different missing rates are considered in different datasets, why not consider 10%, 30%, 50%, 70%, 80%, 90% for them consistently.

W6. Since all the imputation baselines can impute incomplete time series, it is suggested to consider the comparison with them in all datasets, instead of using a subset of them in different datasets.

W7. Scalability experiments are necessary to demonstrate the linear complexity of this paper. Therefore, it is suggested to conduct this on additional large real-world datasets.

W8. There is no parameter sensitivity analysis in experiments, which is also suggested to show the stability of models and illustrate how to determine parameters.

W9. Although the source code has been released, there is not guideline for how to use the codes and how to get the experimental results for each table and figure, leading to the doubts of reproducibility.

**Questions:**

See W1-W9.

---

> ### Author Response · Authors · 2024-11-24
> **Response to Reviewer tnPK (Part I)**
>
> We appreciate Reviewer tnPK for the detailed and constructive comments. We are grateful to the reviewers for their valuable suggestions and thorough review and we have revised our paper accordingly.
>
> **W1:** As discussed in Section 1, the main contribution is proposing a diffusion-based framework for time series imputation, by integrating Mamba-based blocks into diffusion models. However, diffusion models have been proved to be effective in time series imputation, and Mamba is also an existing technique. Modeling bidirectional dependencies and channel dependencies are common techniques to capture characteristics of time series data. Therefore, the novelty and contributions are limited.
>
> **A1:** Thanks a lot for raising this valuable comment.
>
> We would like to argue that our paper is not a simply combination of diffusion model and Mamba, but we propose a Mamba-based structure that can better capture the temporal correlation and channel dependency in multivariate time series with higher effiency than other models.
>
> - To reduce the space and time complexity, we propose Bi-directional Attention Mamba block to capture the sequential correlation (temporal correlation inside the whole time series) instead of Transformer-like structure.
> - To capture channel dependencies, we propose Channel Mamba Block among different channels in a multivariate time series.
>
> With these two mamba-based blocks, we aim to better modeling the multivariate time series even if there are some missing values.
>
> Finally, we use the diffusion model to do time series imputation with the proposed Mamba blocks.
>
>
> **Q2:** There is no theoretical result to support the proposed techniques.
>
> **A2:** Thanks for your valuable suggestion. We feel sorry and are unsure what kind of theoretical results are required. Can you be more specific such that we can make our best efforts in solving it.

---

> ### Author Response · Authors · 2024-11-24
> **Response to Reviewer tnPK (Part II)**
>
> **Q3:** It is unbelievable that the time cost is totally ignored in experiment results, since the main contribution claimed by authors is that the linear complexity of proposed models.
>
> **A3:** Thank you for your comment on time results. We feel very sorry for not providing adequant time results and we have updated more inference time results and comparisons with other diffusion models in **Table 8 in the revised paper** and we also list it here below:
>
> We can see from the table that CSDI has the lowest inference time because it contains the fewest parameter.
> Comparing with SSSD (diffusion-based time series imputation model), the inference time and memory cost of our model is almost the same if we modify our models to make them contains nearly same number of parameters. Secondly, we can see that inference time and gpu memory cost is linear with respect to the number of channels.
>
> |                      | electricity | electricity | electricity    | electricity    | electricity     | MuJoCo | MuJoCo | MuJoCo         | MuJoCo         | MuJoCo          |
> |----------------------|-------------|-------------|----------------|----------------|-----------------|--------|--------|----------------|----------------|-----------------|
> |                      | CSDI        | SSSD        | DiffImp (C=64) | DiffImp (C=96) | DiffImp (C=128) | CSDI   | SSSD   | DiffImp (C=64) | DiffImp (C=96) | DiffImp (C=128) |
> | Model size (M)       | 0.05        | 49.23       | 24.21          | 51.03          | 87.7            | 0.05   | 48.3   | 24             | 50.92          | 87.57           |
> | Inference time(s)    | 0.10        | 0.42        | 0.268          | 0.548          | 0.936           | 0.051  | 0.416  | 0.264          | 0.543          | 0.936           |
> | GPU Memory Cost (MB) | 4046        | 2534        | 1662           | 2724           | 4604            | 3226   | 2448   | 1748           | 2696           | 4574            |

---

> ### Author Response · Authors · 2024-11-24
> **Response to Reviewer tnPK (Part III)**
>
> **Q4:** Only synthetic missing values are considered in experiments. It is suggested to use real-world incomplete datasets to demonstrate the effectiveness and efficiency of proposed methods.
>
> **A4:** Thank you for your valuable feedback on datasets. Here we provide supplmentary experiment results on real-world incomplete datasets (Air quality dataset and healthcare dataset, in **Table 11,12,13 in the revised paper**). We also list the experiment results here below.
> This table shows the MAE and MSE results on Air quality dataset, including more time series imputation baselines. It can be observed that on the Air quality dataset, our method achieves the best MAE performance, with a performance improvement of 30% compared to the second best method. As for the MSE metric, out method get the second best result and is only 1.85% higher than the best result.
>
> |         | AQI       |AQI                   |
> |---------|----------------|-------------------|
> |         | MAE            | MSE               |
> | V-RIN   | 25.4 $\pm$ 0.62  | -                 |
> | GP-VAE  | 25.71          | 2589.53           |
> | BRITS   | 14.1$\pm$ 0.26  | 495.94$\pm$ 43.56  |
> | SPIN    | 11.77$\pm$ 0.54 | -                 |
> | SPIN-H  | 10.89$\pm$ 0.27 | -                 |
> | gatgpt  | 10.28          | **341.26**            |
> | GRIN    | 10.51$\pm$ 0.28 | 371.47$\pm$ 17.38  |
> | CSDI    | 9.60$\pm$ 0.04  | -                 |
> | DiffImp | **6.75$\pm$ 0.014** | 347.58$\pm$ 0.55    |
>
> '-' denotes the corresponding result is not provided in the original paper.
>
> The following table shows the RMSE results on Air quality and Physionet (healthcare) dataset. On the Physionet dataset, our method achieves SOTA RMSE performance on all three missing ratios, with improvement of 22.6%, 17.2%, 23.5% correspondingly. On the Air quality dataset, our method achieves the second best RMSE result. The MSE, MAE and RMSE results demonstrate the effectiveness of our models on time series imputation problems.
>
> |         | Physionet         |        Physionet           |        Physionet           | AQI         |
> |---------|-------------------|-------------------|-------------------|-----------------|
> |         | 10% missing       | 50% missing       | 90% missing       |                 |
> | V-RIN   | 0.628$\pm$ 0.025  | 0.693$\pm$ 0.022  | 0.928$\pm$ 0.013  | 40.11$\pm$ 1.14 |
> | BRITS   | 0.619$\pm$  0.018  |  0.701$\pm$ 0.021 | 0.847$\pm$ 0.021  | 24.28$\pm$ 0.65 |
> | SSGAN   | 0.607$\pm$ 0.034  | 0.758 $\pm$ 0.025 | 0.830$\pm$ 0.009  | -               |
> | RDIS    | 0.635$\pm$ 0.018  | 0.747 $\pm$ 0.013 | 0.922$\pm$ 0.018  | 37.25$\pm$ 0.31 |
> | CSDI    | 0.531$\pm$ 0.009  | 0.668$\pm$ 0.007  | 0.834$\pm$ 0.006  | 19.21$\pm$ 0.13 |
> | CSBI    | 0.547$\pm$ 0.019  | 0.649 $\pm$ 0.009 | 0.837$\pm$ 0.012  | 19.07$\pm$ 0.18 |
> | SSSD    | 0.459$\pm$ 0.001  | 0.632$\pm$ 0.004  | 0.824$\pm$ 0.003  | 18.77$\pm$ 0.08 |
> | TS-Diff | 0.523$\pm$ 0.015  | 0.679$\pm$ 0.009  | 0.845$\pm$ 0.007  | 19.06$\pm$ 0.14 |
> | SAITS   | 0.461$\pm$ 0.009  | 0.636$\pm$ 0.005  | 0.819$\pm$ 0.002  | 18.68$\pm$ 0.13 |
> | D^3M    | 0.438$\pm$ 0.003  | 0.615$\pm$ 0.012  | 0.814$\pm$ 0.002  | **18.19$\pm$ 0.18** |
> | TIDER   | 0.486$\pm$ 0.006  | 0.659$\pm$ 0.009  | 0.833$\pm$ 0.005  | 18.94$\pm$ 0.21 |
> | DiffImp | **0.339$\pm$ 0.0002** | **0.509$\pm$ 0.007**  | **0.623$\pm$ 0.0001** | 18.66$\pm$ 0.26 |
>
> The following table exhibits the CRPS results on Air quality and Physionet (healthcare) dataset. Our method performs SOTA on 10%, 50% missing of Physionet dataset and AQI dataset. The CRPS results indicates that our method has strong capability of handling probabilistic time series imputation problem.
>
> |         | Physionet         |     Physionet     |    Physionet           | AQI             |
> |---------|-------------------|---------------------|-------------------|--------------------|
> |         | 10% missing       | 50% missing         | 90% missing       |                    |
> | GP-VAE  | 0.582$\pm$ 0.003       | 0.796$\pm$ 0.004         | 0.998$\pm$ 0.001       | 0.402$\pm$ 0.009        |
> | V-RIN   | 0.814$\pm$ 0.004       |  0.845$\pm$ 0.002        | 0.932$\pm$ 0.001       | 0.534$\pm$ 0.013        |
> | CSDI    | 0.242$\pm$ 0.001       | 0.336$\pm$ 0.002         | 0.528$\pm$ 0.003       | 0.108$\pm$ 0.001        |
> | CSBI    | 0.247$\pm$ 0.003       | 0.332 $\pm$ 0.003        | 0.527$\pm$ 0.006       | 0.110$\pm$ 0.002        |
> | SSSD    | 0.233$\pm$ 0.001       | 0.331$\pm$ 0.002         | 0.522$\pm$ 0.002       | 0.107$\pm$ 0.001        |
> | TS-Diff | 0.249$\pm$ 0.002       | 0.348$\pm$ 0.004         | 0.541$\pm$ 0.006       | 0.118$\pm$ 0.003        |
> | D^3M    | 0.223$\pm$ 0.001       | 0.327$\pm$ 0.003         | **0.520$\pm$ 0.001**       | 0.106$\pm$ 0.002        |
> | DiffImp | **0.164$\pm$ 0.0004** | **0.2438$\pm$ 0.00008** | 0.533$\pm$ 0.0004 | **0.0959$\pm$ 0.0002** |

---

> ### Author Response · Authors · 2024-11-24
> **Response to Reviewer tnPK (Part IV)**
>
> **W5:** 70%, 80%, 90% missing values are injected into the MuJoCo dataset, and 10%, 30%, 50% missing values are considered into the Electricity dataset. It is wondered why different missing rates are considered in different datasets, why not consider 10%, 30%, 50%, 70%, 80%, 90% for them consistently.
>
> **A5:** Thank you for your insight comments on experiments. First, we choose 10%, 30%, 50% missing ratio for the Electricity dataset and 70%, 80%, 90% for the MuJoCo dataset to be consistent with the experimental settings in [1]. Second, the 10%, 30%, 50%  experiments of MuJoCo dataset and the 70%, 80%, 90% experiments of Electricity dataset are still in progress. We will update in the revised paper as soon as possible.
>
> [1] Juan Miguel Lopez Alcaraz and Nils Strodthoff. Diffusion-based time series imputation and forecasting with structured state space models.
>
> **W6:** Since all the imputation baselines can impute incomplete time series, it is suggested to consider the comparison with them in all datasets, instead of using a subset of them in different datasets.
>
> **A6:** Thank you for your suggestions on experiments. The experiments of comparsion on all datasets are still in progress. We will update in the revised paper as soon as possible.
>
> **W7:** Scalability experiments are necessary to demonstrate the linear complexity of this paper. Therefore, it is suggested to conduct this on additional large real-world datasets.
>
> **A7:** Thank you for your advise on scalability experiments. First, we conduct experiments on electricity and MuJoCo dataset and compare the results with other linear-complexity diffusion models [1], which indicates the linear time and space complexity of our proposed method, the result in presented in **Table 7 in the revised paper**. Second, experiments on additional large real-world datasets is still in progress, we will update in the revised paper as soon as possible.

---

> ### Author Response · Authors · 2024-11-24
> **Response to Reviewer tnPK (Part V)**
>
> **W8:** There is no parameter sensitivity analysis in experiments, which is also suggested to show the stability of models and illustrate how to determine parameters.
>
> **A8:** Thank you for your comment. We agree with the reviewers that a parameter sensitivity analysis is necessary to illustrate the choice of parameters and stability of models. We have provided the parameter sensitivity results in **Table 7 in the revised paper** and we also list it here:
>
> |       | MAE               | MSE                | MRE               | RMSE             |
> |-------|-------------------|--------------------|-------------------|------------------|
> | C=32  | 0.0482$\pm$  0.0004  | 0.0066$\pm$  0.0004   | 0.0496$\pm$  0.00131 | 0.0809$\pm$  0.0025 |
> | C=64  | 0.0147$\pm$  0.00030 | 0.00075$\pm$  0.00007 | 0.0151$\pm$  0.00031 | 0.0273$\pm$  0.0012 |
> | C=128 | 0.0135$\pm$  0.00075 | 0.00065$\pm$  0.00001 | 0.0139$\pm$  0.00076 | 0.0254$\pm$  0.0020 |
>
> In our experiment, we choose C=128 for the following reasons: 1. comparing with C=32, there is a significant performance improvement for larger C 2. comparing with C=64, the performance improvement of setting C = 128 is minor, so we infer that making C larger than 128 may not bring more significant performance improvement. 3. When setting C = 256, we encounter out of gpu memory error. To achieve a balance between effiency and performance, we decide to set C = 128.
>
> **W9:** Although the source code has been released, there is not guideline for how to use the codes and how to get the experimental results for each table and figure, leading to the doubts of reproducibility.
>
> **A9:** Thank you for your suggestion on reproducibility. We feel very sorry for not updating the README file promptly. We have updated the README file in the link provided in **Section 6 in the revised paper**.

---

> > ### Author Response · Authors · 2024-11-26
> > **Waiting for response**
> >
> > Dear Reviewer tnPK,
> >
> > Since the End of the Rebuttal comes soon, we would like to inquire if our response addresses your primary concerns. If it does, we kindly request that you reconsider the score. If you have any additional suggestions, we are more than willing to engage in further discussions and make necessary improvements to the paper. Thanks again for dedicating your time to enhancing our paper!
> >
> > Kind Regards, Authors

---

> > > ### Comment · Reviewer_tnPK · 2024-11-26
> > > **Thanks for responses**
> > >
> > > Thank you for your responses. Since I am waiting for the claimed ``still in progress'' experiments in ``A5, A6, A7'', I did not reply earlier. Depressingly, I have not received the reminder that those ``still in progress'' experiments are finished. In addition, as indicated in W3, the main contribution claimed by authors is that the linear complexity of proposed models. It is surprising that, in the additional time cost experiments, the proposed model DiffImp (C=128) is much slower than the other diffusion-based models, with larger Model sizes and GPU Memory Costs. As claimed by the authors, C=128 is chosen by the authors. Therefore, such results cannot make me accept the contribution of this paper, and I will keep my score 3: reject, not good enough.

---

> > > > ### Author Response · Authors · 2024-11-27
> > > > **Response to Reviewer tnPK (Updating, Part I)**
> > > >
> > > > **W5:** 70%, 80%, 90% missing values are injected into the MuJoCo dataset, and 10%, 30%, 50% missing values are considered into the Electricity dataset. It is wondered why different missing rates are considered in different datasets, why not consider 10%, 30%, 50%, 70%, 80%, 90% for them consistently.
> > > >
> > > > **A5:** Thank you for your suggestions on missing ratios. Firstly, the missing ratios are chosen according to other commonly-used baselines including [1,2,3,4,5] for a fair comparsion. Secondly, we also conduct experiments on 10%, 30%, 50% for MuJoCo and 70%, 80%, 90% for Electricity dataset and the result is in **Table 18** in the revised paper (also list below). Thirdly, let's analyze from the properties of the datasets. MuJoCo is a dataset with low dimension (14) and short sequence length (100), therefore, low missing ratio will easily lead to overfitting problems. Electricity is of very high dimension(370) and the train set is very small (817 sequences), so high missing ratios may leading to very high bias and bad performance. Thus, we consider 10%, 30% and 50% for Electricity and 70%, 80%, 90% for MuJoCo.
> > > >
> > > > |                                        |        MuJoCo       |                                      |                                      |    Electricity    |                                     |                                     |
> > > > |-----------------------------------------|:-------------------------------------:|:------------------------------------:|--------------------------------------|:-----------------------------------:|:-----------------------------------:|-------------------------------------|
> > > > |                                         |         10%        |        30%        |        50%        |        70%       |        80%       |        90%       |
> > > > | MSE                   | 0.0003$\pm$ 0.00001 | 0.0003$\pm$ 0.00001 | 0.0004$\pm$ 0.00001 |  1.469$\pm$ 0.0076 | 2.0085$\pm$ 0.0092 | 3.9443$\pm$ 0.0225 |
> > > > | RMSE                  |  0.0003$\pm$ 0.0004 | 0.0182$\pm$ 0.00015 | 0.0187$\pm$ 0.0001  |  1.212$\pm$ 0.0031 | 1.4167$\pm$ 0.0030 | 1.9856$\pm$ 0.0058 |
> > > > | MAE |  0.0145$\pm$ 0.0002  |  0.0150$\pm$ 0.0001 | 0.0151$\pm$ 0.0001  | 0.7847$\pm$ 0.0013 | 0.9469$\pm$ 0.0023 | 1.3685$\pm$ 0.0033 |
> > > > | MRE |  0.0149$\pm$ 0.0002  |  0.0154$\pm$ 0.0001 | 0.0155$\pm$ 0.0001  | 0.4195$\pm$ 0.0006 | 0.5060$\pm$ 0.0002 | 0.7333$\pm$ 0.0007 |
> > > >
> > > > [1] Che, Z., Purushotham, S., Cho, K., Sontag, D., & Liu, Y. (2018). Recurrent neural networks for multivariate time series with missing values.  *Scientific reports* ,  *8* (1), 6085.
> > > >
> > > > [2] Rubanova, Y., Chen, R. T., & Duvenaud, D. K. (2019). Latent ordinary differential equations for irregularly-sampled time series.  *Advances in neural information processing systems* ,  *32* .
> > > >
> > > > [3]Shan, S., Li, Y., & Oliva, J. B. (2023, June). Nrtsi: Non-recurrent time series imputation. In *ICASSP 2023-2023 IEEE International Conference on Acoustics, Speech and Signal Processing (ICASSP)* (pp. 1-5). IEEE.
> > > >
> > > > [4]Du, W., Côté, D., & Liu, Y. (2023). Saits: Self-attention-based imputation for time series.  *Expert Systems with Applications* ,  *219* , 119619.
> > > >
> > > > [5] Alcaraz, J. M. L., & Strodthoff, N. (2022). Diffusion-based time series imputation and forecasting with structured state space models.  *arXiv preprint arXiv:2208.09399* .

---

> > > > ### Author Response · Authors · 2024-11-27
> > > > **Response to Reviewer tnPK (Updating, Part II)**
> > > >
> > > > **W6.** Since all the imputation baselines can impute incomplete time series, it is suggested to consider the comparison with them in all datasets, instead of using a subset of them in different datasets.
> > > >
> > > > **A6:** Thank you for your suggestions on datasets. Firstly, we carefully reviewed the commonly used datasets in time series imputation task. And we conduct additional experiments on Air quality and healthcare with all baselines. The result is shown in **Table 12,13** in the revised paper (also list below). We can see our DiffImp achieve SOTA results in 6 out of 8 experiments. Besides, the CRPS result contains fewer baselines as it is a metric for evaluating probablistic time series imputation methods and thus we do not calculate the results of determinstic methods.
> > > >
> > > > The following table shows the RMSE results on Air quality and Physionet (healthcare) dataset.
> > > >
> > > > |         | Physionet         |        Physionet           |        Physionet           | AQI         |
> > > > |---------|-------------------|-------------------|-------------------|-----------------|
> > > > |         | 10% missing       | 50% missing       | 90% missing       |                 |
> > > > | V-RIN   | 0.628$\pm$ 0.025  | 0.693$\pm$ 0.022  | 0.928$\pm$ 0.013  | 40.11$\pm$ 1.14 |
> > > > | BRITS   | 0.619$\pm$  0.018  |  0.701$\pm$ 0.021 | 0.847$\pm$ 0.021  | 24.28$\pm$ 0.65 |
> > > > | SSGAN   | 0.607$\pm$ 0.034  | 0.758 $\pm$ 0.025 | 0.830$\pm$ 0.009  | -               |
> > > > | RDIS    | 0.635$\pm$ 0.018  | 0.747 $\pm$ 0.013 | 0.922$\pm$ 0.018  | 37.25$\pm$ 0.31 |
> > > > | CSDI    | 0.531$\pm$ 0.009  | 0.668$\pm$ 0.007  | 0.834$\pm$ 0.006  | 19.21$\pm$ 0.13 |
> > > > | CSBI    | 0.547$\pm$ 0.019  | 0.649 $\pm$ 0.009 | 0.837$\pm$ 0.012  | 19.07$\pm$ 0.18 |
> > > > | SSSD    | 0.459$\pm$ 0.001  | 0.632$\pm$ 0.004  | 0.824$\pm$ 0.003  | 18.77$\pm$ 0.08 |
> > > > | TS-Diff | 0.523$\pm$ 0.015  | 0.679$\pm$ 0.009  | 0.845$\pm$ 0.007  | 19.06$\pm$ 0.14 |
> > > > | SAITS   | 0.461$\pm$ 0.009  | 0.636$\pm$ 0.005  | 0.819$\pm$ 0.002  | 18.68$\pm$ 0.13 |
> > > > | D^3M    | 0.438$\pm$ 0.003  | 0.615$\pm$ 0.012  | 0.814$\pm$ 0.002  | **18.19$\pm$ 0.18** |
> > > > | TIDER   | 0.486$\pm$ 0.006  | 0.659$\pm$ 0.009  | 0.833$\pm$ 0.005  | 18.94$\pm$ 0.21 |
> > > > | DiffImp | **0.339$\pm$ 0.0002** | **0.509$\pm$ 0.007**  | **0.623$\pm$ 0.0001** | 18.66$\pm$ 0.26 |
> > > >
> > > > The following table exhibits the CRPS results on Air quality and Physionet (healthcare) dataset. The CRPS results indicates that our method has strong capability of handling probabilistic time series imputation problem.
> > > >
> > > > |         | Physionet         |     Physionet     |    Physionet           | AQI             |
> > > > |---------|-------------------|---------------------|-------------------|--------------------|
> > > > |         | 10% missing       | 50% missing         | 90% missing       |                    |
> > > > | GP-VAE  | 0.582$\pm$ 0.003       | 0.796$\pm$ 0.004         | 0.998$\pm$ 0.001       | 0.402$\pm$ 0.009        |
> > > > | V-RIN   | 0.814$\pm$ 0.004       |  0.845$\pm$ 0.002        | 0.932$\pm$ 0.001       | 0.534$\pm$ 0.013        |
> > > > | CSDI    | 0.242$\pm$ 0.001       | 0.336$\pm$ 0.002         | 0.528$\pm$ 0.003       | 0.108$\pm$ 0.001        |
> > > > | CSBI    | 0.247$\pm$ 0.003       | 0.332 $\pm$ 0.003        | 0.527$\pm$ 0.006       | 0.110$\pm$ 0.002        |
> > > > | SSSD    | 0.233$\pm$ 0.001       | 0.331$\pm$ 0.002         | 0.522$\pm$ 0.002       | 0.107$\pm$ 0.001        |
> > > > | TS-Diff | 0.249$\pm$ 0.002       | 0.348$\pm$ 0.004         | 0.541$\pm$ 0.006       | 0.118$\pm$ 0.003        |
> > > > | D^3M    | 0.223$\pm$ 0.001       | 0.327$\pm$ 0.003         | **0.520$\pm$ 0.001**       | 0.106$\pm$ 0.002        |
> > > > | DiffImp | **0.164$\pm$ 0.0004** | **0.2438$\pm$ 0.00008** | 0.533$\pm$ 0.0004 | **0.0959$\pm$ 0.0002** |

---

> > > > ### Author Response · Authors · 2024-11-27
> > > > **Response to Reviewer tnPK (Updating, Part III)**
> > > >
> > > > **W7.** Scalability experiments are necessary to demonstrate the linear complexity of this paper. Therefore, it is suggested to conduct this on additional large real-world datasets.
> > > >
> > > > **A7:** Thank you for your comments on scalability experiments. We have just finished the scalability experiment on larger real-world datasets and we will list and analyze the results.
> > > >
> > > > The time complexity analysis part tells that the model has linear complexity w.r.t the number of channels $C$ and input sequence length $L$. The result of experiments on number of channels is shown in **Table 17 and Figure 6** in the revised paper and we also list it here. From the results in the table, we can see that our model is of linear complexity w.r.t the number of channels $C$
> > > >
> > > > | MuJoCo          | MuJoCo             | Electricity     | Electricity        |
> > > > |-----------------|--------------------|-----------------|--------------------|
> > > > | Num of Channels | Inference time (s) | Num of Channels | Inference time (s) |
> > > > | 32              | 0.13               | 32              | 0.134              |
> > > > | 64              | 0.264              | 64              | 0.268              |
> > > > | 96              | 0.543              | 96              | 0.548              |
> > > > | 128             | 0.936              | 128             | 0.936              |
> > > >
> > > > As for the sequence length, for the relatively small imputation datasets, the sequence length is usually very small and the entire sequence is totally fed into the network as the input. So we test the inference time on a large real-world dataset (ettm1) with different sequence length. The result is shown in **Table 16 and Figure 5** in the revised paper. We also list the table below. From the results and the figure, we can see that the inference  time is linear w.r.t the input sequence length, which proves the linear complexity of our model.
> > > >
> > > > | Sequence Length | Inference time (s) |
> > > > |-----------------|--------------------|
> > > > | 120             | 0.93               |
> > > > | 96              | 0.9252             |
> > > > | 480             | 0.944              |
> > > > | 576             | 0.946              |
> > > > | 1056            | 0.966              |

---

> ### Author Response · Authors · 2024-11-27
> **Response to Reviewer tnPK (Updating, Part IV)**
>
> **W3, W8:** About the choice of $C$
>
> **A:** Thank you for your comments on parameter selection. Briefly, setting C = 128 is aiming at achieving a balance between performance and training effiency. We choose C = 128 for the following reasons:
>
> 1. Our model is capable of achieving nearly same inference time comparing with CSDI when setting C = 32 (0.13s vs 0.10s on electricity) with larger params (7.25M vs 0.05M) and lower memory (1182MB vs 4046MB), this can demostrate the potential of inference effectiveness of our models.
> 2. From the experiment results of setting C = 32 and C = 64 (**Table 7** in the revised paper, also list below), we can see that when training for the same iterations (150000 iters), the performance has a significant increase and when setting C = 128, the performance improvement is not that significant, which indicates setting C > 128 may not bring further performance improvements. Also, when setting C = 256, the training process encouter "CUDA Out of Memory Error" on a single GPU, so we decide not to choose C > 128.
>
> |       | MAE               | MSE                | MRE               | RMSE             |
> |-------|-------------------|--------------------|-------------------|------------------|
> | C=32  | 0.0482$\pm$  0.0004  | 0.0066$\pm$  0.0004   | 0.0496$\pm$  0.00131 | 0.0809$\pm$  0.0025 |
> | C=64  | 0.0147$\pm$  0.00030 | 0.00075$\pm$  0.00007 | 0.0151$\pm$  0.00031 | 0.0273$\pm$  0.0012 |
> | C=128 | 0.0135$\pm$  0.00075 | 0.00065$\pm$  0.00001 | 0.0139$\pm$  0.00076 | 0.0254$\pm$  0.0020 |
>
> 3. Another finding in our experiments is that when setting C = 32 or 64, the training process requires more iterations to achieve similar results compared with C = 128 (**Table 15 in the appendix** and is also listed below). So, for training efficiency and the balance between performance and training cost, we finally choose C = 128 in our models.
>
>  |                      |  MSE                 | RMSE              | MAE               | MRE               |
> |----------------------|----------------------|-------------------|-------------------|-------------------|
> |                      |          90%         |       90%             |        90%            |            90%        |
> | C = 64 (300000iter)  | 0.0008$\pm$ 0.00008  | 0.0277$\pm$ 0.0014 | 0.0126$\pm$ 0.0005 | 0.0121$\pm$ 0.0001 |
> | C = 128 (150000iter) | 0.0004$\pm$ 0.00001  | 0.0191$\pm$ 0.0003 | 0.0142$\pm$ 0.0002 | 0.0146$\pm$ 0.0002 |
> |                      |          80%         |      80%              |       80%              |         80%            |
> | C = 64 (300000iter)  | 0.00030$\pm$ 0.00002 | 0.0174$\pm$ 0.0005 | 0.0104$\pm$ 0.0001 | 0.0107$\pm$ 0.0001 |
> | C = 128 (150000iter) | 0.00031$\pm$ 0.00001  | 0.0178$\pm$ 0.0003 | 0.0114$\pm$ 0.0001 | 0.0117$\pm$ 0.0001 |
> |                      |          70%         |        70%           |    70%               |       70%            |
> | C = 64 (300000iter)  | 0.0003$\pm$ 0.00001  | 0.0166$\pm$ 0.0004 | 0.0117$\pm$ 0.0001 | 0.0121$\pm$ 0.0001 |
> | C = 128 (150000iter) | 0.0004$\pm$ 0.00001  | 0.0191$\pm$ 0.0003 | 0.0142$\pm$ 0.0002 | 0.0146$\pm$ 0.0002 |

---

### Official Review · Reviewer_pCfq · 2024-11-05

**Soundness:** 2
**Presentation:** 2
**Contribution:** 1
**Rating:** 3
**Confidence:** 4

**Summary:**

This paper proposes a time series imputation method using Mamba state space model as the backbone of diffusion model. The SSM-based blocks aim to improve inter-variable and bidirectional dependency modeling. Experimental results show this approach achieves state-of-the-art performance across multiple datasets and varying missing scenarios and ratios.

**Strengths:**

1) The proposed method focuses on modeling channel dependencies in time series and the the backbone selection in diffusion models. This is a novel and interesting idea.
2) The conditional diffusion model should allow exploitation of useful information in the data for accurate reconstruction. This is a strength.
3) There are good experimental evaluations and comparisons presented in the paper.

**Weaknesses:**

1) The main motivation of model compatibility in diffusion-based imputation is not well-stated. Seems like authors would investigate which backbone is more suitable for diffusion model. It does not clear what certain limitation in using other models as the backbone, since CNN, Transformer, and SSM are channel independent and unidirectional.
2) Authors claim the proposed method is channel-dependent and has low complexity. But it seems like using SSM in a different way, i.e., modeling across channel correlation. I do not agree this point can be stated as a drawback. If capturing inter-channel relationships is one of the key motivations for this paper, I would like to know why these relationships are essential for the imputation task and what issues might arise from ignoring them. But, this discussion is missing from the paper.
3) In the original SSSD paper, ECG data was also used for experiments, and the model was evaluated based on imputation accuracy and visualization. The ECG experiments were significant in the SSSD paper, yet it appears the authors did not include this experiment in their current paper. Given that the authors replicated other experiments from the SSSD paper, omitting this key experiment seems unusual. Without it, the authors only tested their method on two datasets for the imputation task. Including additional datasets could enhance the reliability of the results.
4) Have the authors tried other approaches for modeling channel-wise dependencies? As noted in the paper, there is no clear order among channels in multivariate time series, so using an order-agnostic attention mechanism for inter-channel relationships is intuitive. However, the motivation for using the Mamba model to capture inter-channel relationships is unclear, as state space models typically assume a temporal order in the data. Could the authors provide further explanation to clarify the rationale behind the channel Mamba block? Additionally, an ablation study on channel modeling would be beneficial.

**Questions:**

Please refer to the weakness. Here my main concern is with the motivation of this paper. It appears to be an exploratory study investigating a good model combination without clearly defining a technical, theoretical, or practical research question. Vanilla CNN, Transformer, and SSM are just model architectures; using them in an appropriate way does not, in itself, demonstrate significance. Overall, the contribution of this paper is limited, and a more comprehensive literature review and clearer positioning within the research field are needed.

---

> ### Author Response · Authors · 2024-11-24
> **Response to Reviewer pCfq (Part I)**
>
> We would like to thank Reviewer pCfq for the valuable comments and insightful feedback of our work and we are thankful for the detailed comments on motivation,experiments and model details.
>
> **W1:** The main motivation of model compatibility in diffusion-based imputation is not well-stated. Seems like authors would investigate which backbone is more suitable for diffusion model. It does not clear what certain limitation in using other models as the backbone, since CNN, Transformer, and SSM are channel independent and unidirectional.
>
> **A1:** Thank you for your comments on movtiation. Our motivation is to investigate **efficient** and **suitable** backbones for diffusion-based probabilistic time series imputation. However, existing diffusion-based models use CNN, Transformer, and SSM as the denoising backbones, as shown in **Table 1** of the revised paper (also list below).
>
> | Backbone Model | Global Dependency |   Time Complexity  | Channel Dependency | Inter-sequence Dependency |
> |----------------|:-----------------:|:------------------:|:------------------:|:-------------------------:|
> | CNN            |       Local       |  $\mathcal{O}(L)$  |     Independent    |       Unidirectional      |
> | Transformer    |       Global      | $\mathcal{O}(L^2)$ |     Independent    |       Unidirectional      |
> | SSM            |      Partial      |  $\mathcal{O}(L)$  |     Independent    |       Unidirectional      |
> | DiffImp~(Ours) |       Global      |  $\mathcal{O}(L)$  |      Dependent     |       Bidirectional       |
>
> From the table, we can have the following conclusions: 1.A CNN-based backbone can only model the dependencies within a local window of channels, this does not match the charatertics of time series data.
> 2.the Transformer backbone has quadratic complexity, which mean it requires high computation cost. 3. Pure SSM based backbones are channel-independent and unidirectional, which does not match the charatertics of time series data.
> Hence, we choose the Mamba model, which can model global relationships and has linear complexity, as the backbone. To model channel dependencies and bidirectional relationships, we make structural modifications to Mamba, which are Bidirectional Attenion Mamba for temporal correlations and Channel Mamba block for channel dependencies.

---

> ### Author Response · Authors · 2024-11-24
> **Response to Reviewer pCfq (Part II)**
>
> **W2:** Authors claim the proposed method is channel-dependent and has low complexity. But it seems like using SSM in a different way, i.e., modeling across channel correlation. I do not agree this point can be stated as a drawback. If capturing inter-channel relationships is one of the key motivations for this paper, I would like to know why these relationships are essential for the imputation task and what issues might arise from ignoring them. But, this discussion is missing from the paper.
>
> **A2:** Thank you for your insightful comments on model details of channel dependency modeling.
> First, there exisits channel dependencies in time series data and in the time series imputation problem, there may be missing points in both sides of observed points. Theoretically, if we abandon modeling channel dependencies and bidirectional relationships in time series, the model's accuracy in modeling the overall joint distribution will decrease, leading to biases in both deterministic and uncertainty estimations. Experimentally, we conducted **ablation studies (Table 9 in the revised paper)** on cases where only the forward sequence was modeled, only the backward sequence was modeled, and where channel dependencies were removed. The result is listed in the following table. We can see that the removal (or modification) of bidirectional modeling and channel dependency modeling leads to significant performance drops on all metrics.
>
> | Time Modeling | Temporal Attention | Inter-Channel Dependency |         MSE         |         MAE         |         MRE        |         RMSE        |
> |:-------------:|:------------------:|:------------------------:|:-------------------:|:-------------------:|:------------------:|:-------------------:|
> | Bidirectional |         Yes        |            Yes           | 5.46e-4$\pm$ 1.6e-5 | 1.17e-2$\pm$ 7.4e-5 | 1.21$\pm$ 7.5e-3\% | 2.33e-2$\pm$ 3.1e-4 |
> |    Forward    |         Yes        |            Yes           | 7.19e-4$\pm$ 2.0e-5 | 1.26e-2$\pm$ 2.1e-4 | 1.29$\pm$ 2.2e-2\% | 2.67e-2$\pm$ 2.9e-4 |
> |    Forward    |         Yes        |            No            | 7.48e-4$\pm$ 9.5e-5 | 1.23e-2$\pm$ 3.5e-4 | 1.23$\pm$ 3.5e-2\% | 2.71e-2$\pm$ 1.5e-3 |
> |    Backward   |         Yes        |            Yes           | 7.24e-4$\pm$ 7.3e-5 | 1.30e-2$\pm$ 4.2e-4 | 1.30$\pm$ 4.2e-2\% | 2.69e-2$\pm$ 1.3e-3 |
> |    Backward   |         Yes        |            No            | 8.39e-4$\pm$ 6.1e-5 | 1.46e-2$\pm$ 3.8e-4 | 1.46$\pm$ 3.8e-2\% | 2.89e-2$\pm$ 1.0e-3 |
> | Bidirectional |         Yes        |            No            | 8.85e-4$\pm$ 2.8e-5 | 1.40e-2$\pm$ 3.1e-4 | 1.44$\pm$ 3.4e-2\% | 2.97e-2$\pm$ 4.8e-4 |
> | Bidirectional |         No         |            Yes           | 9.66e-4$\pm$ 9.5e-5 | 1.53e-2$\pm$ 3.3e-4 | 1.57$\pm$ 3.5e-2\% | 3.09e-2$\pm$ 1.3e-3 |
> | Bidirectional |         Yes        |     Channel Attention    | 7.43e-4$\pm$ 4.0e-5 | 1.31e-2$\pm$ 5.5e-5 | 1.35$\pm$ 5.6e-3\% | 2.71e-2$\pm$ 5.6e-5 |

---

> ### Author Response · Authors · 2024-11-24
> **Response to Reviewer pCfq (Part III)**
>
> **W3:**  In the original SSSD paper, ECG data was also used for experiments, and the model was evaluated based on imputation accuracy and visualization. The ECG experiments were significant in the SSSD paper, yet it appears the authors did not include this experiment in their current paper. Given that the authors replicated other experiments from the SSSD paper, omitting this key experiment seems unusual. Without it, the authors only tested their method on two datasets for the imputation task. Including additional datasets could enhance the reliability of the results.
>
> **A3:** Thank you for your insightful comments on baselines and datasets. We agree with the reviewer that the experiments on ECG dataset(PTB-XL dataset) should be included in our paper for reliability of the results. We have made the following revisions according to your comment:
>
> 1. We add the experiment results on ECG dataset (**Table 14 in the appendix of the revised paper**) and we also list the experiment results here. We can see from the following table that our model outperforms all other baselines on all settings of ECG data imputation.
>    | Model     | MAE     | RMSE    |
> |--------------------|------------------|------------------|
> | 20\% RM on PTB-XL  |                  |                  |
> | LAMC               | 0.0678           | 0.1309           |
> | CSDI               | 0.0038$\pm$ 2e-6      | 0.0189$\pm$ 5e-5      |
> | DiffWave           | 0.0043$\pm$ 4e-4      | 0.0177$\pm$ 4e-4      |
> | SSSD               |**0.0034$\pm$ 4e-6**     | 0.0119$\pm$ 1e-4      |
> | DiffImp            | **0.0034$\pm$ 2e-5**     | **0.0101$\pm$ 3e-4** |
> | 20\% RBM on PTB-XL |                  |                  |
> | LAMC               | 0.0759           | 0.1498           |
> | CSDI               | 0.0186$\pm$ 1e-5      | 0.0435$\pm$ 2e-4      |
> | DiffWave           | 0.0250$\pm$ 1e-3      | 0.0808$\pm$ 5e-3      |
> | SSSD               | 0.0103$\pm$ 3e-3      | 0.0226$\pm$ 9e-4      |
> | DiffImp            | **0.0067$\pm$ 3e-5** | **0.0221$\pm$ 1e-3** |
> | 20\% BM on PTB-XL  |                  |                  |
> | LAMC               | 0.0840           | 0.1171           |
> | CSDI               | 0.1054$\pm$ 4e-5      | 0.2254$\pm$ 7e-5      |
> | DiffWave           | 0.0451$\pm$ 7e-4      | 0.1378$\pm$ 5e-3      |
> | SSSD               | 0.0324$\pm$ 3e-3      | 0.0832$\pm$ 8e-3      |
> | DiffImp            | **0.022 $\pm$ 4e-5**   | **0.059$\pm$ 1e-3**    |
> 2. We add experiements on PhysioNet dataset and air quality dataset in **Table 11,12,13 in the revised paper**. And we also show the supplementary experimental results here.
>    This table shows the MAE and MSE results on Air quality dataset, including more time series imputation baselines. It can be observed that on the Air quality dataset, our method achieves the best MAE performance, with a performance improvement of 30% compared to the second best method. As for the MSE metric, out method get the second best result and is only 1.85% higher than the best result.
>
> |         | AQI       |AQI                   |
> |---------|----------------|-------------------|
> |         | MAE            | MSE               |
> | V-RIN   | 25.4 $\pm$ 0.62  | -                 |
> | GP-VAE  | 25.71          | 2589.53           |
> | BRITS   | 14.1$\pm$ 0.26  | 495.94$\pm$ 43.56  |
> | SPIN    | 11.77$\pm$ 0.54 | -                 |
> | SPIN-H  | 10.89$\pm$ 0.27 | -                 |
> | gatgpt  | 10.28          | **341.26**            |
> | GRIN    | 10.51$\pm$ 0.28 | 371.47$\pm$ 17.38  |
> | CSDI    | 9.60$\pm$ 0.04  | -                 |
> | DiffImp | **6.75$\pm$ 0.014** | 347.58$\pm$ 0.55    |
>
> '-' denotes the corresponding result is not provided in the original paper.
>
> See Response Part IV for more experiment results.

---

> ### Author Response · Authors · 2024-11-24
> **Response to Reviewer pCfq (Part IV)**
>
> This part continues from response part III.
>
> The following table shows the RMSE results on Air quality and Physionet (healthcare) dataset. On the Physionet dataset, our method achieves SOTA RMSE performance on all three missing ratios, with improvement of 22.6%, 17.2%, 23.5% correspondingly. On the Air quality dataset, our method achieves the second best RMSE result. The MSE, MAE and RMSE results demonstrate the effectiveness of our models on time series imputation problems.
>
> |         | Physionet         |        Physionet           |        Physionet           | AQI         |
> |---------|-------------------|-------------------|-------------------|-----------------|
> |         | 10% missing       | 50% missing       | 90% missing       |                 |
> | V-RIN   | 0.628$\pm$ 0.025  | 0.693$\pm$ 0.022  | 0.928$\pm$ 0.013  | 40.11$\pm$ 1.14 |
> | BRITS   | 0.619$\pm$  0.018  |  0.701$\pm$ 0.021 | 0.847$\pm$ 0.021  | 24.28$\pm$ 0.65 |
> | SSGAN   | 0.607$\pm$ 0.034  | 0.758 $\pm$ 0.025 | 0.830$\pm$ 0.009  | -               |
> | RDIS    | 0.635$\pm$ 0.018  | 0.747 $\pm$ 0.013 | 0.922$\pm$ 0.018  | 37.25$\pm$ 0.31 |
> | CSDI    | 0.531$\pm$ 0.009  | 0.668$\pm$ 0.007  | 0.834$\pm$ 0.006  | 19.21$\pm$ 0.13 |
> | CSBI    | 0.547$\pm$ 0.019  | 0.649 $\pm$ 0.009 | 0.837$\pm$ 0.012  | 19.07$\pm$ 0.18 |
> | SSSD    | 0.459$\pm$ 0.001  | 0.632$\pm$ 0.004  | 0.824$\pm$ 0.003  | 18.77$\pm$ 0.08 |
> | TS-Diff | 0.523$\pm$ 0.015  | 0.679$\pm$ 0.009  | 0.845$\pm$ 0.007  | 19.06$\pm$ 0.14 |
> | SAITS   | 0.461$\pm$ 0.009  | 0.636$\pm$ 0.005  | 0.819$\pm$ 0.002  | 18.68$\pm$ 0.13 |
> | D^3M    | 0.438$\pm$ 0.003  | 0.615$\pm$ 0.012  | 0.814$\pm$ 0.002  | **18.19$\pm$ 0.18** |
> | TIDER   | 0.486$\pm$ 0.006  | 0.659$\pm$ 0.009  | 0.833$\pm$ 0.005  | 18.94$\pm$ 0.21 |
> | DiffImp | **0.339$\pm$ 0.0002** | **0.509$\pm$ 0.007**  | **0.623$\pm$ 0.0001** | 18.66$\pm$ 0.26 |
>
> The following table exhibits the CRPS results on Air quality and Physionet (healthcare) dataset. Our method performs SOTA on 10%, 50% missing of Physionet dataset and AQI dataset. The CRPS results indicates that our method has strong capability of handling probabilistic time series imputation problem.
>
> |         | Physionet         |     Physionet     |    Physionet           | AQI             |
> |---------|-------------------|---------------------|-------------------|--------------------|
> |         | 10% missing       | 50% missing         | 90% missing       |                    |
> | GP-VAE  | 0.582$\pm$ 0.003       | 0.796$\pm$ 0.004         | 0.998$\pm$ 0.001       | 0.402$\pm$ 0.009        |
> | V-RIN   | 0.814$\pm$ 0.004       |  0.845$\pm$ 0.002        | 0.932$\pm$ 0.001       | 0.534$\pm$ 0.013        |
> | CSDI    | 0.242$\pm$ 0.001       | 0.336$\pm$ 0.002         | 0.528$\pm$ 0.003       | 0.108$\pm$ 0.001        |
> | CSBI    | 0.247$\pm$ 0.003       | 0.332 $\pm$ 0.003        | 0.527$\pm$ 0.006       | 0.110$\pm$ 0.002        |
> | SSSD    | 0.233$\pm$ 0.001       | 0.331$\pm$ 0.002         | 0.522$\pm$ 0.002       | 0.107$\pm$ 0.001        |
> | TS-Diff | 0.249$\pm$ 0.002       | 0.348$\pm$ 0.004         | 0.541$\pm$ 0.006       | 0.118$\pm$ 0.003        |
> | D^3M    | 0.223$\pm$ 0.001       | 0.327$\pm$ 0.003         | **0.520$\pm$ 0.001**       | 0.106$\pm$ 0.002        |
> | DiffImp | **0.164$\pm$ 0.0004** | **0.2438$\pm$ 0.00008** | 0.533$\pm$ 0.0004 | **0.0959$\pm$ 0.0002**

---

> ### Author Response · Authors · 2024-11-24
> **Response to Reviewer pCfq (Part V)**
>
> **Q4:** Have the authors tried other approaches for modeling channel-wise dependencies? As noted in the paper, there is no clear order among channels in multivariate time series, so using an order-agnostic attention mechanism for inter-channel relationships is intuitive. However, the motivation for using the Mamba model to capture inter-channel relationships is unclear, as state space models typically assume a temporal order in the data. Could the authors provide further explanation to clarify the rationale behind the channel Mamba block? Additionally, an ablation study on channel modeling would be beneficial.
>
> **A4:** Thank you for your insight comments on channel dependency modeling. The Channel Mamba block is designed according to the following:
>
> Firstly, we propose an attention structure to capture the channel dependency among multivariate time series. As shown in  **Fig.3C** , in the CMB module, we first flip the input sequence $X \in \mathbb{R}^{B \times C \times L}$, then feed it through a linear projection module, followed by a sigmoid module. This process results in a weight tensor with the same dimensions as the input, with values in the range of $[-1, 1]$. This weight tensor can be regarded as the attention weight of the input, and thus, our CMB module includes an attention structure for distributing weights across different channels.
>
> Secondly, we propose mamba block to model the sequential dependencies among channels. For example, in the Mujoco dataset, different channels represent the positional relationships in the motion simulation process. Due to the inherent correlations in robot movements, the channels have latent sequential dependencies. In the AQI dataset, different channels represent data collected from different sensors, which are influenced by real-world factors (such as wind speed). The latent relations (i.e latent sequential dependencies and real-world factors) indicate that the channel data also contains latent sequential dependencies. Therefore, in addition to the attention structure, we also propose a channel Mamba block.
>
> Furthermore, for the channel Mamba block, we conduct ablation experiments. We perform ablation experiments on the channel attention module, where the attention mechanism was implemented using SENet[1]. The experimental results are shown in **Table 11 in the revised paper**  and we also list it below.
> We can see from the following table that the variant with Channel Attention (last line in the table) performs worse on all metrics than the original model (first line in the table). The result proves the necessity of channel mamba blocks.
>
> |          Time Modeling          |   Temporal Attention  |       Inter-Channel Dependency      |                  MSE                 |                  MAE                 |                 MRE                 |                 RMSE                 |
> |:-------------------------------:|:---------------------:|:-----------------------------------:|:------------------------------------:|:------------------------------------:|:-----------------------------------:|:------------------------------------:|
> |          Bidirectional          |          Yes          |                 Yes     | **5.46e-4$\pm$ 1.6e-5** | **1.17e-2$\pm$ 7.4e-5** | **1.21$\pm$ 7.5e-3\%**                   | **2.33e-2$\pm$ 3.1e-4**    |
> |             Forward             |          Yes          |                 Yes   | 7.19e-4$\pm$ 2.0e-5         | 1.26e-2$\pm$ 2.1e-4                   | 1.29$\pm$ 2.2e-2\%                   | 2.67e-2$\pm$ 2.9e-4         |
> |             Forward             |          Yes          |   No     | 7.48e-4$\pm$ 9.5e-5   | 1.23e-2$\pm$ 3.5e-4         | 1.23$\pm$ 3.5e-2\%         | 2.71e-2$\pm$ 1.5e-3                   |
> |             Backward            |          Yes          |                 Yes     | 7.24e-4$\pm$ 7.3e-5   | 1.30e-2$\pm$ 4.2e-4                   | 1.30$\pm$ 4.2e-2\%                   | 2.69e-2$\pm$ 1.3e-3   |
> |             Backward            |          Yes          |                  No                 | 8.39e-4$\pm$ 6.1e-5                   | 1.46e-2$\pm$ 3.8e-4                   | 1.46$\pm$ 3.8e-2\%                   | 2.89e-2$\pm$ 1.0e-3   |
> |          Bidirectional          |          Yes          |                  No                 | 8.85e-4$\pm$ 2.8e-5                   | 1.40e-2$\pm$ 3.1e-4                   | 1.44$\pm$ 3.4e-2\%                   | 2.97e-2$\pm$ 4.8e-4      |
> |          Bidirectional          |           No          |                 Yes                 | 9.66e-4$\pm$ 9.5e-5                   | 1.53e-2$\pm$ 3.3e-4                   | 1.57$\pm$ 3.5e-2\%                   | 3.09e-2$\pm$ 1.3e-3    |
> | Bidirectional | Yes | Channel Attention | 7.43e-4$\pm$ 4.0e-5 | 1.31e-2$\pm$ 5.5e-5 | 1.35$\pm$ 5.6e-3\% | 2.71e-2$\pm$ 5.6e-5 |
>
> [1] Hu, Jie, Li Shen, and Gang Sun. "Squeeze-and-excitation networks."  *Proceedings of the IEEE conference on computer vision and pattern recognition* . 2018.

---

> > ### Author Response · Authors · 2024-11-26
> > **Waiting for response**
> >
> > Dear Reviewer pCfq,
> >
> > Since the End of the Rebuttal comes soon, we would like to inquire if our response addresses your primary concerns. If it does, we kindly request that you reconsider the score. If you have any additional suggestions, we are more than willing to engage in further discussions and make necessary improvements to the paper. Thanks again for dedicating your time to enhancing our paper!
> >
> > Kind Regards, Authors

---

### Official Review · Reviewer_2h3v · 2024-11-06

**Soundness:** 3
**Presentation:** 3
**Contribution:** 2
**Rating:** 5
**Confidence:** 4

**Summary:**

In this paper, the authors investigate the problem of time series imputation. Since the authors find that the DDPM-based models are usually not capable of achieving sequence modeling with low time complexity and cannot handle the inter-variables and temporal dependence. The authors integrate Mamba and develop the bidirectional model for time series imputation and forecasting. The authors evaluate the proposed method on several datasets and achieve good performance.

**Strengths:**

N.A.

**Weaknesses:**

1.	The contribution is limited. This paper looks like a combination of the diffusion model and Mamba. Specifically, the advantage of low time complexity comes from Mamba, and the technique of changing the SSM-based block to a bidirectional one is not very new. A similar idea has been used in bi-LSTM.
2.	In addition, this article mainly solves the imputation problem, but the author considers time series forecasting, which makes the whole work seem to be pieced together.
3.	For fair comparison, authors should consider common imputation benchmarks such as the healthcare dataset in PhysioNet Challenge 2012 and air quality dataset.
4.	Although the author mainly emphasizes that this method is better than other methods based on the diffusion model, this is not enough. The author should consider more time series imputation methods such as [1,2,3,4].


[1] Filling the Gaps: Multivariate Time Series Imputation by Graph Neural Networks.
[2] Gatgpt: A pre-trained large language model with graph attention network for spatiotemporal imputation.
[3] Learning to Reconstruct Missing Data from Spatiotemporal Graphs with Sparse Observations.
[4] Multi-Variate Time Series Forecasting on Variable Subsets.

**Questions:**

N.A.

---

> ### Author Response · Authors · 2024-11-24
> **Response to Reviewer 2h3v (Part I)**
>
> We would like to sincerely thank Reviewer 2h3v for providing a detailed review and insightful comments regarding motivation and contribution, important basic baselines, more datasets, and details about time series and forecasting. We have revised our paper accordingly.
>
> **W1:** The contribution is limited. This paper looks like a combination of the diffusion model and Mamba. Specifically, the advantage of low time complexity comes from Mamba, and the technique of changing the SSM-based block to a bidirectional one is not very new. A similar idea has been used in bi-LSTM.
>
> **A1:** Thanks a lot for raising this valuable comment.
>
> We would like to argue that our paper is not a simply combination of diffusion model and Mamba, but we propose a Mamba-based structure that can better capture the temporal correlation and channel dependency in multivariate time series with higher effiency than other models.
>
> - To reduce the space and time complexity, we propose Bi-directional Attention Mamba block to capture the sequential correlation (temporal correlation inside the whole time series) instead of Transformer-like structure.
> - To capture channel dependencies, we propose Channel Mamba Block among different channels in a multivariate time series.
>
> With these two mamba-based blocks, we aim to better modeling the multivariate time series even if there are some missing values.
>
> Finally, we use the diffusion model to do time series imputation with the proposed Mamba blocks.
>
> **W2:** In addition, this article mainly solves the imputation problem, but the author considers time series forecasting, which makes the whole work seem to be pieced together.
>
> **A2:** Thank you for your insight feedback about our paper.
> According to the definition of time series imputation problem, the time series forecasting problem can be viewed as one case of time series imputation. As stated in the paper (in  **Sec 3.1**), there may be different types of missing patterns in time series imputation. Time series forecasting can be seen as a special case of the imputation problem, where missing values are all located at the end of the sequence. Therefore, models designed for time series imputation are also suitable for time series forecasting tasks. In our paper, we evaluate the performance of the model on time series forecasting tasks to demonstrate its effectiveness.

---

> ### Author Response · Authors · 2024-11-24
> **Response to Reviewer 2h3v (Part II)**
>
> **W3 and W4:**
>
> **W3:** For fair comparison, authors should consider common imputation benchmarks such as the healthcare dataset in PhysioNet Challenge 2012 and air quality dataset.
>
> **W4:** Although the author mainly emphasizes that this method is better than other methods based on the diffusion model, this is not enough. The author should consider more time series imputation methods such as [1,2,3,4].
>
> **A3 and A4:**
> Thank you for your insightful suggestions on baselines and datasets. We agree with you and have conducted more experiments to verify the effectiveness of our methods by including more baselines [1,2,3] and datasets (PhysioNet and Air Quality). The modifications are listed as follows.
>
> 1. We add experiements on PhysioNet dataset and air quality dataset in **Table 11,12,13 in the revised paper**.
> 2. We add comparisons with more time series imputation methods in **Table 11 in the revised paper**.
>
> And we also show the supplementary experimental results here.
> This table shows the MAE and MSE results on Air quality dataset, including more time series imputation baselines. It can be observed that on the Air quality dataset, our method achieves the best MAE performance, with a performance improvement of 30% compared to the second best method. As for the MSE metric, out method get the second best result and is only 1.85% higher than the best result.
> |         | AQI       |AQI                   |
> |---------|----------------|-------------------|
> |         | MAE            | MSE               |
> | V-RIN   | 25.4 $\pm$ 0.62  | -                 |
> | GP-VAE  | 25.71          | 2589.53           |
> | BRITS   | 14.1$\pm$ 0.26  | 495.94$\pm$ 43.56  |
> | SPIN    | 11.77$\pm$ 0.54 | -                 |
> | SPIN-H  | 10.89$\pm$ 0.27 | -                 |
> | gatgpt  | 10.28          | **341.26**            |
> | GRIN    | 10.51$\pm$ 0.28 | 371.47$\pm$ 17.38  |
> | CSDI    | 9.60$\pm$ 0.04  | -                 |
> | DiffImp | **6.75$\pm$ 0.014** | 347.58$\pm$ 0.55    |
>
> '-' denotes the corresponding result is not provided in the original paper.
>
>
> The following table shows the RMSE results on Air quality and Physionet (healthcare) dataset. On the Physionet dataset, our method achieves SOTA RMSE performance on all three missing ratios, with improvement of 22.6%, 17.2%, 23.5% correspondingly. On the Air quality dataset, our method achieves the second best RMSE result. The MSE, MAE and RMSE results demonstrate the effectiveness of our models on time series imputation problems.
>
> |         | Physionet         |        Physionet           |        Physionet           | AQI         |
> |---------|-------------------|-------------------|-------------------|-----------------|
> |         | 10% missing       | 50% missing       | 90% missing       |                 |
> | V-RIN   | 0.628$\pm$ 0.025  | 0.693$\pm$ 0.022  | 0.928$\pm$ 0.013  | 40.11$\pm$ 1.14 |
> | BRITS   | 0.619$\pm$  0.018  |  0.701$\pm$ 0.021 | 0.847$\pm$ 0.021  | 24.28$\pm$ 0.65 |
> | SSGAN   | 0.607$\pm$ 0.034  | 0.758 $\pm$ 0.025 | 0.830$\pm$ 0.009  | -               |
> | RDIS    | 0.635$\pm$ 0.018  | 0.747 $\pm$ 0.013 | 0.922$\pm$ 0.018  | 37.25$\pm$ 0.31 |
> | CSDI    | 0.531$\pm$ 0.009  | 0.668$\pm$ 0.007  | 0.834$\pm$ 0.006  | 19.21$\pm$ 0.13 |
> | CSBI    | 0.547$\pm$ 0.019  | 0.649 $\pm$ 0.009 | 0.837$\pm$ 0.012  | 19.07$\pm$ 0.18 |
> | SSSD    | 0.459$\pm$ 0.001  | 0.632$\pm$ 0.004  | 0.824$\pm$ 0.003  | 18.77$\pm$ 0.08 |
> | TS-Diff | 0.523$\pm$ 0.015  | 0.679$\pm$ 0.009  | 0.845$\pm$ 0.007  | 19.06$\pm$ 0.14 |
> | SAITS   | 0.461$\pm$ 0.009  | 0.636$\pm$ 0.005  | 0.819$\pm$ 0.002  | 18.68$\pm$ 0.13 |
> | D^3M    | 0.438$\pm$ 0.003  | 0.615$\pm$ 0.012  | 0.814$\pm$ 0.002  | **18.19$\pm$ 0.18** |
> | TIDER   | 0.486$\pm$ 0.006  | 0.659$\pm$ 0.009  | 0.833$\pm$ 0.005  | 18.94$\pm$ 0.21 |
> | DiffImp | **0.339$\pm$ 0.0002** | **0.509$\pm$ 0.007**  | **0.623$\pm$ 0.0001** | 18.66$\pm$ 0.26 |
>
> See Respose Part III for more experiment results

---

> ### Author Response · Authors · 2024-11-24
> **Response to Reviewer 2h3v (Part III)**
>
> This part continues from response part II.
>
> **A4**:
> The following table exhibits the CRPS results on Air quality and Physionet (healthcare) dataset. Our method performs SOTA on 10%, 50% missing of Physionet dataset and AQI dataset. The CRPS results indicates that our method has strong capability of handling probabilistic time series imputation problem.
>
> |         | Physionet         |     Physionet     |    Physionet           | AQI             |
> |---------|-------------------|---------------------|-------------------|--------------------|
> |         | 10% missing       | 50% missing         | 90% missing       |                    |
> | GP-VAE  | 0.582$\pm$ 0.003       | 0.796$\pm$ 0.004         | 0.998$\pm$ 0.001       | 0.402$\pm$ 0.009        |
> | V-RIN   | 0.814$\pm$ 0.004       |  0.845$\pm$ 0.002        | 0.932$\pm$ 0.001       | 0.534$\pm$ 0.013        |
> | CSDI    | 0.242$\pm$ 0.001       | 0.336$\pm$ 0.002         | 0.528$\pm$ 0.003       | 0.108$\pm$ 0.001        |
> | CSBI    | 0.247$\pm$ 0.003       | 0.332 $\pm$ 0.003        | 0.527$\pm$ 0.006       | 0.110$\pm$ 0.002        |
> | SSSD    | 0.233$\pm$ 0.001       | 0.331$\pm$ 0.002         | 0.522$\pm$ 0.002       | 0.107$\pm$ 0.001        |
> | TS-Diff | 0.249$\pm$ 0.002       | 0.348$\pm$ 0.004         | 0.541$\pm$ 0.006       | 0.118$\pm$ 0.003        |
> | D^3M    | 0.223$\pm$ 0.001       | 0.327$\pm$ 0.003         | **0.520$\pm$ 0.001**       | 0.106$\pm$ 0.002        |
> | DiffImp | **0.164$\pm$ 0.0004** | **0.2438$\pm$ 0.00008** | 0.533$\pm$ 0.0004 | **0.0959$\pm$ 0.0002** |
>
> [1] Filling the Gaps: Multivariate Time Series Imputation by Graph Neural Networks.
>
> [2] Gatgpt: A pre-trained large language model with graph attention network for spatiotemporal imputation.
>
> [3] Learning to Reconstruct Missing Data from Spatiotemporal Graphs with Sparse Observations.

---

> > ### Author Response · Authors · 2024-11-26
> > **Waiting for response**
> >
> > Dear Reviewer 2h3v,
> >
> > Since the End of the Rebuttal comes soon, we would like to inquire if our response addresses your primary concerns. If it does, we kindly request that you reconsider the score. If you have any additional suggestions, we are more than willing to engage in further discussions and make necessary improvements to the paper. Thanks again for dedicating your time to enhancing our paper!
> >
> > Kind Regards, Authors

---

> > > ### Comment · Reviewer_2h3v · 2024-11-28
> > >
> > > Thanks for your response! Although the reviewer provided an explanation, it did not directly answer my question. For me, the contribution is so incremental, mainly because there are many classic works that process time series from two directions. In addition, the advantages of the proposed method claimed by the author mainly come from Mamba. In addition, Section 3.1 focuses on diffusion models for time series imputation rather than missing patterns. I admit that time series forecasting is a special case of imputation, but they are indeed two different problems and should be treated differently. For example, for time series forecasting, we should study how to utilize periodicity and non-stationarity, etc better. For time series imputation, we should emphasize how to better utilize context, how to deal with various missing patterns, and the assumptions corresponding to different missing patterns. If the author can focus on one problem, it will help improve this paper.

---

### Author Response · Authors · 2024-11-24
**Summary of Changes**

We would like to thank all the reviewers for the time and effort they dedicated to reviewing our paper. We have received many constructive comments and suggestions. Based on these we have revised our paper. All updates are highlighted in blue in the revised paper and the main updates are summarized as follows:

1. In the section of Introduction, we rewrite the parts of model compatibility and contributions for more clear expression.
2. We add supplementary experimental results on the PTB-XL (ECG data), Physionet, and Healthcare datasets, as well as additional comparisons with non-diffusion baselines to demonstrate the effectiveness of our approach.
3. We add experiments results on gpu memory cost, inference time and model parameters to validate the efficiency of our method.
4. We add ablation results for the channel mamba module.
5. We include a parameter sensitivity experiment to explain our parameter choices.
6. We correct some writing typos, grammar and spelling mistakes.

---

### Meta-Review · Area_Chair_iS8M · 2024-12-20

**Metareview:**

This paper proposes a novel approach to time series imputation by integrating the Mamba state space model into a diffusion-based framework. The integration of Mamba for low time complexity and the design of bidirectional modeling for inter-variable dependencies is novel. However, the contribution is considered limited, and the motivation for using Mamba for channel-wise dependencies is unclear.

**Additional Comments On Reviewer Discussion:**

Although the paper has some merits, such as the integration of Mamba for low time complexity and the design of bidirectional modeling for inter-variable dependencies, the issues raised by the reviews are critical. For instance, the contribution is limited, as the paper appears to be a combination of existing techniques (2h3v), the motivation for using Mamba for channel-wise dependencies is unclear (pCfq), and the novelty and theoretical support for the proposed method is insufficient (tnPK). Although the authors address some of the issues in their response, the response did not fully convince the reviewers and the paper still needs a major revision before it can be accepted, especially in addressing the lack of clarity in the motivation, the need for more comprehensive experiments and theoretical analysis, and the unresolved concerns about the model's efficiency.

---

### Decision · Program_Chairs · 2025-01-22

Reject